



# Impacts of Regional-transported Biomass Burning Emissions on Chemical and Optical Properties of Carbonaceous Aerosols in Nanjing, East China

Xiaoyan Liu[1,2,3,4], Yan-Lin Zhang[1,2,3], Yiran Peng[5,6], Lulu Xu[5,6], Chunmao Zhu[7], Fang Cao[1,2,3], Xiaoyao Zhai[1,2,3], Mozammel Haque[1,2,3], Chi Yang[1,2,3], Yunhua Chang[1,2,3], Tong Huang[1,2,3], Zufei Xu[1,2,3], Mengying Bao[1,2,3], Wenqi Zhang[1,2,3], Meiyi Fan[1,2,3], Xuhui Lee[1,2,3,4]

[1]Yale-NUIST Center on Atmospheric Environment, Joint International Research Laboratory of Climate and Environment Change (ILCEC), Nanjing University of Information Science and Technology, Nanjing 210044, China
[2]Key Laboratory of Meteorological Disaster Ministry of Education (KLME), Collaborative Innovation Center on Forecast and Evaluation of Meteorological Disasters (CIC-FEMD), Nanjing University of Information Science and Technology, Nanjing 210044, China
[3]School of Applied Meteorology, Nanjing University of Information Science and Technology, Nanjing 210044, China
[4]School of Forestry and Environmental Studies, Yale University, New Haven 06511, USA
[5]Ministry of Education Key Laboratory for Earth System Modeling, Department of Earth System Science, Tsinghua University, Beijing 100084, China
[6]Joint Center for Global Change Studies (JCGCS), Beijing 100084, China
[7]Research and Development Center for Global Change, Japan Agency for Marine-Earth Science and Technology, Yokohama 236-0001, Japan

*Correspondence* : Yan-Lin Zhang (dryanlinzhang@outlook.com, zhangyanlin@nuist.edu.cn)

**Abstract.** Biomass burning can significantly impact the chemical and optical properties of carbonaceous aerosols. Here, the impacts of biomass burning emissions on chemical and optical properties of carbonaceous aerosols were studied during wintertime in a megacity of Nanjing, East China. The high abundance of biomass burning tracers such as levoglucosan (lev), mannosan (man), galactosan (gal) and non-sea-salt potassium (nss-K$^+$) was found during the studied period with the concentration ranges of 22.4-1476 ng m$^{-3}$, 2.1-56.2 ng m$^{-3}$, 1.4-32.2 ng m$^{-3}$, and 0.2-3.8 μg m$^{-3}$, respectively. Backward air mass origin analysis, potential emission sensitivity of element carbon (EC), and MODIS fire spot information indicated that the elevations of the carbonaceous aerosols were due to the transported biomass-burning aerosols from southeast China. The characteristic mass ratio maps of lev/man and lev/nss-K$^+$ suggested that the biomass fuels were mainly crop residuals. Furthermore, the strong correlation (p < 0.01) between biomass burning tracers (e.g. lev) and light absorption coefficient (b$_{abs}$) for water soluble organic carbon (WSOC) revealed that biomass burning emissions played a significant role in the light-absorption properties of carbonaceous aerosols. The solar energy absorption due to water-soluble brown carbon (BrC) and EC was estimated by the calculation-based on measured light-absorbing parameters and the simulation-based on a radiative transfer model (RRTMG_SW). The solar energy absorption of water-soluble BrC in short wavelengths (300-400 nm) was 0.8 ± 0.4 (0.2-2.3) W m$^{-2}$ from the calculation-based and 1.2 ± 0.5 (0.3-1.9) W m$^{-2}$ from the RRTMG_SW model. The absorption capacity of water-soluble BrC accounted for about 20-30% of the total absorption of EC aerosols. The solar energy absorption



of WSOC due to biomass burning was estimated as 0.2 ± 0.1 (0.0-0.9) W m$^{-2}$. Potential Source Contribution Function model simulations showed that the solar energy absorption induced by WSOC and EC aerosols was mostly due to the regional transported carbonaceous aerosols from the source regions such as southeast China. Our results illustrate the importance of the absorbing water-soluble brown carbon aerosols in trapping additional solar energy in the low-level atmosphere, heating the surface and inhibiting the energy from escaping the atmosphere. The regional transported biomass burning emissions may significantly impact the chemical and optical properties of carbonaceous aerosols in the polluted atmosphere.

# 1 Introduction

Biomass burning is of great concern in recent years due to its severe impact on air quality and climate (Zhang et al., 2015; Gilman et al., 2015; Chen et al., 2017b). Water-soluble organic carbon (WSOC) from biomass burning emission has a pronounced influence on the increase of aerosol cloud condensation nuclei (CCN) activity, which can lead to a cooling impact (Gao et al., 2003; Rogers et al., 1991; Novakov and Corrigan, 1996). Meanwhile, a portion of organic carbon (OC) could yield a cooling effect on the land surface by scattering sunlight (Zhang et al., 2017; Myhre et al., 2013). A portion of OC involves in the aging process of elemental carbon (EC) or black carbon (BC) as coating material for the BC core, enhancing BC radiative absorption (Peng et al., 2016; Wang et al., 2018b). A fraction of OC can also directly absorb solar energy functioning similarly as BC, which is referred to as brown carbon (BrC). But BrC is different from BC due to its strong light absorption from visible (VIS) to ultraviolet (UV) wavelengths (Andreae and Gelencsér, 2006). Radiative forcing of BC is estimated to be 0.2-1.2 W m$^{-2}$ (Moffet and Prather, 2009), while radiative forcing of BrC contribution is suggested to be up to 0.25 W m$^{-2}$ or to contribute to 19 % of the total atmospheric absorption computed by model simulations (Feng et al., 2013). As a part of WSOC, water-soluble BrC are rich in the biomass burning smoke (Washenfelder et al., 2015). Taken together, biomass burning can affect climate and aerosol chemical compositions in an extremely complex way. All effects mentioned above are both regional and global owing to the aerosols long-range transport. Studies based on simulation modeling have shown that the emission from biomass burning can be transported to remote sites, even across the oceans (Aouizerats et al., 2015; Zhu et al., 2016; Ancellet et al., 2016).

Due to the significant role of carbonaceous aerosols in biomass burning plumes, it is important to quantify the contribution from biomass burning to carbonaceous species such as OC, EC, WSOC, and BrC. In the observation-based studies, the characteristic ratio calculation and positive matrix factorization (PMF) are the most frequently methods biomass burning contribution. OC and EC proportions released by biomass burning were shown to be 45 % ± 12 % and 12 % ± 7.3 %, respectively, in the harvest season in Daejeon, Korea (Jung et al., 2014). These numbers are comparable with the results for the harvest seasons in Yangtze River Delta region, China, where biomass burning contributed 51 % and 16 % to OC and EC concentrations, respectively (Chen et al., 2017a). The share of OC in PM$_{2.5}$ produced by biomass burning in Beijing ranged from 18 % to 70 % in different seasons and areas (Duan et al., 2004; Zhang et al., 2008a; Cheng et al., 2013). In addition, a large quantity of BrC with light-absorbing property was detected in biomass burning emissions in South Asia and the United States, and in some controlled wood pyrolysis experiments (Bosch et al., 2014; Chen and Bond, 2010; Hecobian et al., 2010;



Kirchstetter et al., 2004). It was reported that 23 ± 7 % and 16 ± 7 % of WSOC originated from biomass burning in Beijing, in winter and summer, respectively (Yan et al., 2015). Meanwhile, the contributions of biomass burning to WSOC in $PM_{10}$ in Mexico ranged from 7 % to 57 % (Tzompa-Sosa et al., 2016). These studies have demonstrated that a significant importance of biomass burning contributions to carbonaceous aerosols. Levoglucosan is recommended to be a biomass burning tracer

because it is uniquely derived from cellulose pyrolysis (Simoneit et al., 1999; Simoneit, 2002; Simoneit et al., 2004). Levoglucosan has been initially quantified by gas chromatography-mass spectrometry (GC-MS) with a relatively complex operation procedure and at a high cost (Fraser and Lakshmanan, 2000; Graham et al., 2003; Oros and Simoneit, 2001). With the improvement of technology, a new detection method, named high-performance anion-exchange chromatography coupled to pulsed amperometric detection (HPAEC-PAD), has emerged (Engling et al., 2006). The HPAEC-PAD approach is as

reliable as traditional methods (e.g., GC), but with a higher efficiency, an easier operation and a lower cost. The new method can simultaneously quantify other carbohydrates in ambient samples including mannosan and galactosan which are also anhydrosugars and used as tracers of biomass burning like levoglucosan (Engling et al., 2006; Jung et al., 2018; Tsai et al., 2015). Additionally, non-sea-salt potassium ($nss\text{-}K^+$) is also a tracer of biomass burning, calculated by the formula: $nss\text{-}K^+ = K^+ - 0.0335 \times Na^+$, to exclude $K^+$ originated from seawater (Lai et al., 2007; Cao et al., 2016). Nowadays, the above four tracers

are applied to both qualitatively and quantitatively investigate the biomass burning impact. Furthermore, their characteristic ratios help researchers to better recognize the biomass fuel types (Puxbaum et al., 2007; Mochida et al., 2010; Wang et al., 2011; Cheng et al., 2014; Zhang et al., 2012; Cheng et al., 2013)

Biomass burning aerosol has significant impact on regional and global radiative forcing (Ramanathan and Carmichael, 2008). There is no doubt that element carbon increases solar absorption of ambient aerosols most effectively (Ramanathan and

Carmichael, 2008; Myhre and Samset, 2015; Li et al., 2016b). Lack et al. (2012) found that the absorptivity of BC and POM mixture has 70% enhancement in biomass burning period than normal days in Boulder, Colorado.  However, the role of BrC in the radiation balance of the climate system has attracted increasing interest. Shamjad et al. (2015) mentioned that the atmospheric radiative forcing of total aerosol increases by 56%, and the forcing of BrC aerosol increases by 38% in the biomass burning season comparing to other seasons in Kanpur, India. Light absorption characteristics of WSOC aerosols are key factors

in the climate forcing calculation. Light-absorbing parameters including absorption coefficient, mass absorption efficiency and absorption Angstrom exponent, have been involved to describe the light absorption characteristics of BrC released by biomass burning (Alexander et al., 2008; Yang et al., 2009; Chakrabarty et al., 2016; Choudhary et al., 2017; Srinivas et al., 2016). Hoffer et al. (2006) investigated the optics of HULIS in Amazon basin and found that the contribution of HULIS to total aerosol absorption was as high as 50% at the wavelength of 300 nm. Hecobian et al. (2010) measured the optical properties of

WSOC in Atlanta, US and their results showed that the absorption coefficient around 365 nm had a strong correlation with WSOC from biomass burning or from urban emissions. Du et al. (2014b) calculated mass absorbance efficiency for WSOC from a biomass burning factor derived from positive matrix factorization (PMF) receptor model. It showed that the biomass burning source contributed to 58% of total light absorption at 365 nm and WSOC associated with sulfate and oxalate contributed to 21%. Using a simplistic absorption-based model,  Kirillova et al. (2014b) estimated the relative absorptive

forcing of WSOC compared to EC in winter season at New Delhi, India. Their results indicated that the contribution of WSOC




to total absorption was between 3% and 11%. Li et al. (2016a) applied a similar method in the Tibetan Plateau and found that the relative contributions of radiative forcing of WSOC to EC were $6.03 \pm 3.62\%$ and $11.41 \pm 7.08\%$ at two stations there. The afore-mentioned method provides a rough estimation on the absorption of solar energy by WSOC and EC, by using the measured sample concentrations of WSOC and EC and the optical properties of WSOC ($MAC_{365}$ and AAE). Many

assumptions were applied in that method (e.g., MAC and AAE of EC being constant), thus uncertainties in the estimated forcing could be substantial. Additionally, Jung et al. (2015) evaluated the instantaneous forcing of WSOC. They calculated the aerosol optical properties and applied a simplified expression to account for the radiative transfer process. Their results showed that the radiative forcing of WSOC aerosol (assumed core-shell mixing with inorganic components) range from -0.07 to -0.49 W m$^{-2}$ at the top-of-atmosphere. In a most recent study, Panicker et al. (2017) applied an aerosol optical model

embedded in a radiative transfer model to investigate the radiative forcing for OC and EC under two urban environments in Northern India. The model can produce quite realistic results as long as the default database and user-specified inputs represent the real atmospheric condition in model. In this study, we follow to estimate the radiative forcing of BrC and WSOC with a radiative transfer model.

Physical and optical properties of carbonaceous aerosols alter with the transport process and modify their radiative forcing

accordingly. For example, pollutants from biomass burning in North Asia and North America can transport to Arctic with the poleward jet in mid-latitude and have non-negligible effect on Arctic climate (Shindell and Faluvegi, 2009). During the transport process, BC and BrC are prone to be internally mixed with other gas and particle pollutants from biomass burning, resulting in multiplication of the absorptivity due to the lensing effect (Bond and Bergstrom, 2006). Quantification of BrC property change with the transport of biomass burning aerosol and estimation of its contribution to the regional radiative

forcing are essential for better understanding the climate effect of BrC. So far only a few studies have focused on the light-absorbing property of BrC from biomass burning in China (Cheng et al., 2011; Du et al., 2014b; Yan et al., 2015). It was shown that over short wavelengths (300-400nm), light absorption by WSOC, which is often used as the surrogate for water-soluble BrC, was ~40 % and ~25 % of that by EC in winter and summer, respectively (Yan et al., 2015). However, the BrC light absorption quantification due to biomass burning has not been identified.

China, the top crop-producing country in the world has a history of burning crop-residuals (Zhang et al., 2016), making China a hotspot of biomass burning research. As soon as it became evident that biomass burning emission is detrimental to the environment and human health, China's governments introduced policies to prohibit straw burning. These policies are now implemented strictly, so that open biomass burning has been quite rare in recent years especially in eastern China. Under this background, this study mainly aimed at a wintertime pollution event in Yangtze River Delta region, China and intended to

determine the origin of biomass burning aerosols and figure out the impacts of regional-transported biomass burning on chemical and optical properties of carbonaceous aerosols. To achieve these goals, 3 h $PM_{2.5}$ samples has been collected and analyzed to determine the aerosols chemical compositions and light-absorbing properties. The solar energy absorption of carbonaceous aerosols was estimated by the calculation-based on measured light-absorbing parameters and the simulation-based on a radiative transfer model (RRTMG_SW).



## 2 Method

### 2.1 Site description and sample collection

The observation site NUPT (118.78° E, 32.09° N) was located on the campus of Nanjing University of Posts and Telecommunications. It is in the urban area of eastern China where is one of the China's economic centres and also one of the

5 world's most advanced manufacturing base. This site represents a severely anthropogenically influenced environment, with a complex pollution pattern contributed by vehicles, industries, biomass burning and atmospheric transport.

The sampling campaign was conducted from 14 Jan 2015 to 30 Jan 2015. $PM_{2.5}$ samples were collected at a flow rate of 1.05 $m^3$ $min^{-1}$ with a nominal sampling time of 3 h for each sample. Blank filters were kept in the sampler for 1 min, without air flow, at both the beginning and the end of this campaign. All filters used for sample collection were quartz fiber filters, which

had been prebaked at 450° C for 6 h to eliminate organic material.

### 2.2 Sample analysis for carbonaceous components

OC and EC were detected using a Desert Research Institute (DRI) Model 2001 Thermal Optical Carbon Analyzer (Atmoslytic Inc., Calabasas, USA). A portion of sampled filter (0.53 $cm^2$) was analyzed following the IMPROVE thermal optical reflectance (TOR) protocol (Bao et al., 2017). For the determination of WSOC, a portion of each filter with a size of 4.02 $cm^2$

was removed from the parent filter and extracted with 10 mL ultrapure water in a sonication bath for 30 min. The water extracts were filtered through syringe filters (0.22μm, ANPEL, Shanghai, China) for WSOC concentration analysis by a TOC analyzer (TOC-L, Shimadzu, Kyoto, Japan) following the WSTC-WSIC protocol. All the calculations related to the carbonaceous components were blank corrected and the system error between DRI and TOC-L analyzers was corrected by testing sucrose solution of a specific concentration.

### 2.3 Extraction and analysis of ion and carbohydrates

A portion of each sample filter (5.07 $cm^2$) was cut off and extracted with 10.0 mL ultrapure water (> 18.2Ω) via ultrasonic agitation (30 min). The extract solution was filtered to remove insoluble materials, then used for ion analysis by ion chromatography on a ThermoFisher Scientific ICS-5000+ system (US) equipped with a gradient pump (SP), a conductivity detector/chromatography compartment (DC) and an automated sampler (AS-DV). The separation of cation was carried out on

an IonPac CS12A analytical column and an IonPac CG12A guard column with aqueous methanesulfonic acid (MSA, 30 mM $L^{-1}$) eluent at a flow rate of 1 mL $min^{-1}$. While anions were separated on an IonPac AS11-HC analytical column and an IonPac AG11-HC guard column using sodium hydroxide (NaOH) gradient elution at a flow rate of 1.5 mL $min^{-1}$: 0-3 min, 0.5 mM $L^{-1}$; 3-5 min, 0.5-5 mM $L^{-1}$; 5-15 min, 5-30 mM $L^{-1}$; 15-20 min, 0.5 mM $L^{-1}$.

The extraction procedure for carbohydrate analysis was basically same as for ion, except that water volume was changed from

30 10.0 to 2.0 mL. The equipment of the ion chromatography system used was largely the same, except that a conductivity detector was replaced by an electrochemical detector. The carbohydrates were separated by a CarboPac MA1 column and a matched guard column using NaOH gradient elution at a flow rate of 0.4 mL $min^{-1}$. To determine levoglucosan and galactosan the



elution was run in a time-gradient: -15-34 min, 300 mM L$^{-1}$; 34-45 min, 300-480 mM L$^{-1}$; 45-60 min, 480 mM L$^{-1}$. For mannosan, the elution was run in another time-gradient: -15-5 min, 50 mM L$^{-1}$; 5-25 min, 250 mM L$^{-1}$; 25-35 min, 250-350 mM L$^{-1}$; 30-40 min, 350-650 mM L$^{-1}$; 40-70 min, 650 mM L$^{-1}$. The volume of Loop was 200 μL. All ions and carbonaceous components amount calculations were corrected by the mean value of two field blanks.

## 2.4 Extraction and analysis of WSOC light-absorbing property

A fraction of sampled filter (5.02 cm$^2$) was cut and extracted with 2 mL ultrapure water. The filter extract was tested for light absorption by an ultraviolet-visible absorption spectrophotometer (UV-2600, Shimadzu, Kyoto, Japan) with a scanning wavelength range of 200-800 nm. Absorption coefficient (b$_{abs}$, M m$^{-1}$ or 10$^{-6}$ m$^{-1}$) at 365 nm was calculated by Eq. (1):

$$b_{abs} = (A_{365} - A_{700}) \times (V_{water} \times Factor) \times ln(10) \div (V_{aero} \times L), \tag{1}$$

where $A_{365}$ and $A_{700}$ refer to absorbance (or light attenuation) at 365 and 700 nm, respectively, measured by the spectrometer; $V_{water}$ (mL) corresponds to the volume of the aqueous extract (water); $Factor$ is set to 103, which is estimated from the absorption signal for the full filter; $V_{aero}$ (m$^3$) refers to the volume of air filtered; and $L$ (mm) is the path length of the cell (10mm). Choice of 365 nm as the reference wavelength to represent light absorption of WSOC is made according to the strong light-absorbing capacity and also is meant to avoid light-absorbing disturbance by other substances in the extract. Absorbance at 700 nm represents the baseline drift in the analysis (Bosch et al., 2014; Cheng et al., 2011; Hecobian et al., 2010). Mass absorption efficiency (MAE, m$^2$ g$^{-1}$) of BrC in WSOC was derived from Eq. (2):

$$MAE = b_{abs}/WSOC, \tag{2}$$

where WSOC refers to the concentration of WSOC (μg m$^{-3}$). Absorption Angstrom exponents (AAE) were computed by Eq. (3):

$$b_{abs} \approx K \cdot \lambda^{-AAE}, \tag{3}$$

where $K$ is a constant, $AAE$ describes the wavelength-dependent absorption enhancement of BrC, associated with its origin, size and composition (Bikkina, 2014); and $\lambda$ is wavelength in the range of 310-460 nm. AAE for WSOC was fitted within the range of 310-460 nm in which coefficients of determination (R$^2$) for all samples are above 0.90. The solar energy absorbance of WSOC and EC at the ground level was estimated using Eq. (4) and Eq. (5):

$$E_{WSOC} = \int I(\lambda) \cdot \{1 - e^{-(MAE \cdot (\frac{365}{\lambda})^{AAE} \cdot WSOC \cdot h_{ABL})}\} d\lambda, \tag{4}$$

$$E_{EC} = \int I(\lambda) \cdot \{1 - e^{-(MAE_{EC} \cdot (\frac{550}{\lambda}) \cdot EC \cdot h_{ABL})}\} d\lambda, \tag{5}$$

where $I(\lambda)$ is the clear sky Air Mass 1 Global Horizontal (AM1GH) solar radiance spectrum at the surface; $h_{ABL}$ refers to vertical height of the boundary layer, (set to 1000m); $MAE_{EC}$ is set as 7.5 ± 1.2m$^2$ g$^{-1}$ which is MAE of EC at 550 nm; $EC$ (μg m$^{-3}$) is the EC concentration in each sample. The fraction $E_{WSOC}/ E_{EC}$ is the ratio of $E_{WSOC}$ and $E_{EC}$ comparing light-absorbing capacity of BrC in WSOC and EC. Here $E_{WSOC}$, $E_{EC}$ and $E_{WSOC}/E_{EC}$ were computed in a wavelength range of 300-400 nm (Bosch et al., 2014; Kirillova et al., 2014b; Bond and Bergstrom, 2006; Yan et al., 2015).



## 2.5 Radiative forcing estimation with model

We also estimated the radiative forcing of WSOC and EC with a stand-alone radiative transfer model RRTMG_SW (Global climate model version of Short Wave Rapid Radiative Transfer Model (Iacono Michael et al., 2000). The correlated-k approach was applied in RRTMG_SW to calculate radiative fluxes and heating rates in 14 contiguous bands in the shortwave (820-

50000 cm$^{-1}$). The stand-alone RRTMG_SW can be driven by a give atmospheric profile. In this study, we prescribed the mid-latitude winter atmospheric profile, which is default in the RRTMG package (Iacono Michael et al., 2008; Mlawer and Clough, 1998). We chose the average solar zenith angle of 60° in the mid-latitude regions (Li, 2017) and ran RRTMG_SW for clear-sky conditions without the impact of cloud.

Aerosol radiative (absorptive) properties for WSOC and EC are parameterized in terms of their mass concentrations in

RRTMG_SW. The optics per unit mass (mass absorption efficiency (MAE$^*$), single scattering albedo (SSA$^*$), and asymmetry factor (g$^*$)) of EC was provided from the Optical Properties of Aerosols and Clouds (OPAC) dataset (Hess et al., 1998). For WSOC, we calculated the optics using the Mie model (Bohren and Huffman, 1998; Pruppacher and Klett, 1997) with the input optics and prescribed size parameters. Refractive Index (RI) was set to be 1.55-0.112i (Kirchstetter et al., 2004; Shamjad et al., 2016). The density of WSOC is 1.569 g cm$^{-3}$ (Feng et al., 2013). The dry mode radius and standard deviation of the WSOC

particle size distribution were assumed to be 0.0212 μm and 2.24, respectively (Hess et al., 1998). Because WSOC is hydrophilic, we also calculated the wet particle radius and wet RIs according to (Pruppacher and Klett, 1997). The optics of WSOC obtained from the Mie model were provided to RRTMG_SW for radiative calculations.

## 2.6 Footprint analysis, PSCF models and fire hotspots

The Flexpart Lagrangian particle dispersion model was used to estimate the footprint of the site (Stohl et al., 1998; Stohl et

al., 2005; Grythe et al., 2017). Flexpart version 10.1 was run in the backward model in which the potential emission sensitivity (PES) of the receptor point is provided (Seibert and Frank, 2004). The operational reanalysed data from the European Centre for Medium-Range Weather Forecasts (ECMWF) at a spatial resolution of 1° × 1° with 61 vertical levels were used as meteorology. The simulation considered BC (corresponding to EC in this study) as the tracer species, and dry deposition and wet scavenging were accounted for. The output was set as the retention time(s) of particles in each grid during the simulation

period. HYSPLIT 4.8 model from NOAA was used to compute 48 h backward trajectories of air masses reaching Nanjing sites at a planetary boundary layer (PBL) height of 500 m. MODIS fire hotspot data was applied to evaluate open biomass burning intensity during the study period. The Potential Source Contribution Function model (PSCF) was usually applied to localize the potential sources of pollutants. The details about the setup of the model can be seen in  the research of Bao et al. (2017). Here we introduce the light absorbance of WSOC into the model. The higher the PSCF value is, the higher possibility the areas

make potential contributions to the light absorbance of WSOC in the aerosols at the receptor site.



## 3. Results and discussion

### 3.1 Carbonaceous components

The temporal variations of OC/EC ratios, and OC and EC concentrations in $PM_{2.5}$ during the studied episode are plotted in Fig. 1. The average concentrations of OC and EC were $19.1 \pm 8.4$ μg m$^{-3}$ and $6.5 \pm 3.4$ μg m$^{-3}$, respectively. Our results are

comparable to the seasonal mean levels of OC and EC level in $PM_{2.5}$ in the winter of 2015 in a nearby site in Nanjing, which were $22.5 \pm 9.6$ and $8.2 \pm 3.1$μg m$^{-3}$, respectively (Li et al., 2015). OC shows a robust relationship with EC ($R^2 = 0.9$, $p < 0.01$), which underlines the significant contribution of fuel burning to OC, since EC is only released by inefficient combustion (Liu et al., 2014a). In this study, EC peaked at 17:00 on 21 Jan, and 20:00 on 24 Jan, implying intensive fuel burning at those times. OC ranged from 5.5 to 45.8 μg m$^{-3}$ and EC ranged from 0.6 to 20.1 μg m$^{-3}$. The relative standard deviation was used

to represent the uncertainty of OC (44%) and EC (53%), suggesting that EC had a larger certainty than OC. It indicates that EC came from a variety of combustion sources which were dynamic and unstable. The variation of OC and EC over time led to the significant changes of $PM_{2.5}$ levels.

The OC/EC ratios varied from 2.0 to 7.2, with a mean value of $3.1 \pm 0.7$. The OC/EC ratio is usually used as the indicator of secondary organic carbon (Hou et al., 2011; Zeng and Wang, 2011). According to the temporal variation of the OC/EC ratio

in Fig. 1, the impact of secondary organic carbon varied in different stage of the pollution episode. The OC/EC ratio variation range was 2.0-7.2 when the atmosphere was cleaner. As the pollution got more severe over time, it shrank to 2.0-4.0 after 16 Jan. In contrast to combustion sources, the secondary sources had a relatively weak and stable impact on $PM_{2.5}$ at the polluted stage.

Temporal variations of WSOC concentration and WSOC/OC ratio are illustrated in Fig. 1. The average concentration of WSOC

in the event was $9.7 \pm 4.3$ μg m$^{-3}$ and WSOC varied from 2.2 to 23.2 μg m$^{-3}$. WSOC was in a higher level in this pollution episode than in the episodes reported by Du et al. (2014a). It was reported that WSOC was abundant in biomass burning emissions in previous studies (Jaffrezo et al., 2005; Park and Cho, 2011). In addition, a strong correlation was also found between OC and WSOC ($R^2 > 0.7$, $p < 0.01$) and WSOC/OC mass ratio was averaged to be 0.5, a value being consistent with that of another study (Bikkina, 2014). If a major portion of OC was derived from combustion, WSOC could be very likely

released from fuel burning. Moreover, WSOC is an important form of BrC, which is able to absorb light radiation. It seems that biomass burning can emit BrC, increasing the light-absorbing capacity of aerosols.

### 3.2 Chemical species related to fossil fuel combustion

To explore the combustion sources, the roles of traffic and industrial emissions, which are main fossil fuel combustion sources in the urban area (Cao et al., 2005), were first investigated. Nitrogen dioxide ($NO_2$) is the main chemical component in motor

vehicle exhaust (Kendrick et al., 2015), and sulfur dioxide ($SO_2$) is usually treated as the tracer for coal combustion widely existing in industrial activities (Akimoto and Narita, 1994). Therefore, the sum of $NO_2$ and $SO_2$ concentration can represent the contribution of fossil fuel combustion to the atmospheric pollutants. As shown in Fig. 2, carbon monoxide (CO) and the sum of $NO_2$ and $SO_2$ concentrations have a positive correlation, but with a low correlation coefficient ($R^2 = 0.40$, $p < 0.01$).



The same pattern is also shown between EC and sum of nitrate ($NO_3^-$) and sulfate ($SO_4^{2-}$) in $PM_{2.5}$ ($R^2 = 0.46$, $p < 0.01$). These patterns suggest that fossil fuel was not the only main type of fuel and that biomass fuel might have been burned during the episode.

### 3.3 Biomass burning tracers

Tracers of biomass burning have been introduced to describe the role of biomass burning in this event. As shown in Fig. 3, all tracers strongly correlate with each other ($R^2 > 0.60$, $p < 0.01$). But there is a more significant correlation between anhydrosugars ($R^2 > 0.75$, $p < 0.01$) than that between nss-$K^+$ and anhydrosugars ($0.61 \leqslant R^2 \leqslant 0.67$, $p < 0.01$). Because of the non-unique origin of potassium, nss-$K^+$ seems a weak tracer of biomass burning, in agreement with other published results. $K^+$ has additional significant sources other than biomass burning, such as seawater, soil resuspension and fertilizers (Urban et

al., 2012). Furthermore, $K^+$ is abundant in firework aerosols according to several recent studies (Cheng et al., 2013; Drewnick et al., 2006). Among all three types of anhydrosugars, levoglucosan was treated as the main tracer in this study because of its larger quantity in $PM_{2.5}$ than the other two.

Time series plots of nss-$K^+$, levoglucosan, mannosan and galactosan in $PM_{2.5}$ are shown in Fig. 1. In this study, nss-$K^+$ varied from 0.2 to 3.8 µg m$^{-3}$, with a mean level of $1.2 \pm 0.7$ µg m$^{-3}$. A consistent trend between nss-$K^+$ and EC can be seen in Fig. 1,

especially with both peaks of EC and nss-$K^+$ during 14:00-17:00 on 21 Jan (15.7 and 3.8 µg m$^{-3}$) and during 17:00-20:00 on 24 Jan (20.1 and 3.5µg m$^{-3}$), suggesting that the intensities of total fuel combustion and biomass fuel combustion changed synchronously.

The average concentration of levoglucosan in $PM_{2.5}$ was $373 \pm 268$ ng m$^{-3}$, which was significantly higher than those observed in other areas, e.g., at coastal sites in Europe (Puxbaum et al., 2007), at forest sites in eastern China (Wang *et al.*, 2008), and

at an urban site in Shanghai, China (Li et al., 2016c), but was lower than those measured at an urban site in Beijing during winter and a biomass burning episode of summer (Cheng et al., 2013)(Table 2).

The concentration range of levoglucosan in $PM_{2.5}$ of our study was 22.4-1476 ng m$^{-3}$. The maximum value approximately approached that observed in Shanghai in the autumn when biomass burning prevailed (1606 ng m$^{-3}$, Table 2). The highest level of levoglucosan appeared during 05:00-08:00 on Jan 25, implying that the strongest biomass fuel combustion occurred in this

period. Remarkably, levoglucosan concentration presented a valley during 11:00-14:00 on 19 Jan, when EC presenting a peak. It can be explained that the intense combustion at that time was not dominated by biomass burning. The discrepancy of time distribution between nss-$K^+$ and levoglucosan were again attributed to the diversity of origins of potassium.

The concentrations of mannosan and galactosan in $PM_{2.5}$ were averaged $18.5 \pm 12.5$ ng m$^{-3}$ and $9.9 \pm 6.7$ ng m$^{-3}$, with ranges of 2.1-56.2 ng m$^{-3}$ and 1.4-32.2 ng m$^{-3}$, respectively. The trend of mannosan, galactosan and levoglucosan were closely

matched, as illustrated in Fig. 1. They reached their highest and second highest levels together, which highlighted the fact that biomass burning heavily impacted the compositions of aerosols in these two periods.



### 3.4 Biomass burning contribution to carbonaceous components

Figure 4 shows the correlations between carbonaceous components (OC, EC or WSOC) and biomass burning tracers (levoglucosan, mannosan, galactosan or nss-$K^+$). All biomass burning tracers are associated with EC ($0.40 < R^2 < 0.80$, $p < 0.01$), which demonstrates the conjecture that biomass burning was one of the main types of combustion. Moreover, biomass burning tracers exhibit strong correlations with OC and WSOC ($0.40 < R^2 < 0.80$, $p < 0.01$), indicating that biomass burning might have contributed to organic carbon including WSOC. For each tracer, the higher coefficient of determination ($R^2$) for OC than EC illustrates that the biomass burning impacts on OC were more significant than those on EC. Notably, the slopes and the intercepts of the trend line for WSOC are both about half of those for OC, in accordance with results shown in Sect. 3.1 that WSOC accounted for around 50 % of OC. As a consequence, biomass burning made a stable contribution to WSOC and the contribution was even larger than that to EC in this episode.

The contribution of biomass burning to OC in $PM_{2.5}$ (BB-OC) was calculated using Eq. (6):

$$BB - OC = \frac{(lev - 17.5)/(1000 \times OC)}{0.082} \times 100\,\%, lev: ng\ m^{-3}; OC: \mu g\ m^{-3}, \tag{6}$$

(Zhang et al., 2010; Puxbaum et al., 2007; Zdráhal et al., 2002). In biomass burning source emission tests for three major types of cereal straw (corn, wheat, and rice) in China, 0.082 was reported as the lev/OC ratio for $PM_{2.5}$ (Zhang et al., 2007). This value can be used in combination with the lev/OC ratios of our $PM_{2.5}$ samples to roughly estimate the contribution of biomass burning smoke to the ambient OC. The reason why we used the ratio lev/OC associated with cereal straw burning will be explained in Sect. 3.5. BB-OC ranged from 0.2 % to 53.6 % in this event, with an average of 20.9 ± 9.3 %. Our calculated biomass burning contribution has a greater span than that of Beijing rural area (18–38 %) which was computed with the same lev/OC ratio (Zhang et al., 2008a; Zhang et al., 2008b), even though January was not in a common biomass burning prevailing season of Nanjing. There were three remarkable BB-OC peaks at 08:00 on 20 Jan, 14:00 on 21 Jan and 08:00 on 25 Jan in Fig. 5, indicating the large contributions of biomass burning during those periods. However, EC and biomass burning tracers were not rich at 08:00 on 20 Jan, implying that combustion activities were not highly intense at that time. But it doesn't contradict the fact that biomass burning made a significant contribution to OC compared to other pollution sources at 08:00 on 20 Jan, which illustrating the necessity of parameter BB-OC.

The contribution from biomass burning to the WSOC in $PM_{2.5}$ (BB-WSOC) trend in Fig. 5 makes the above analysis more convincing due to the consistency of the contributions computed by two different methods. BB-WSOC was estimated with Eq. (7):

$$BB - WSOC = \frac{(lev/WSOC)_{ambient}}{0.17} \times 100\,\%, lev/WSOC: \mu g\ \mu g C^{-1}, \tag{7}$$

which used a lev/WSOC ratio of 0.17 μg μgC⁻¹ from the test burns of rice straws and wheat straw (Yan et al., 2015). BB-WSOC ranged from 1.1 % to 55.4 %, with an average of 22.3 ± 9.9 %. It was at an equivalent level with that investigated in Beijing wintertime where BB-WSOC was averaged at 23 ± 7 % (Yan et al., 2015), and had a pretty large span. BB-WSOC



exhibits a similar trend with BB-OC (Fig. 5). The robust relationship between BB-OC and BB-WSOC ($R^2 = 0.81$, $p < 0.01$) with a regression slope close to 1 (0.96) confirms the reliability of our biomass burning contribution quantification.

### 3.5 Origin of biomass burning

### 3.5.1 Major fuel types of biomass burning

The biomass burning characteristics, expressed in the parameter space of lev/man and lev/nss-$K^+$ in Fig. 6, are used to differentiate the burning substrates. The ratio space for needle, duff, hardwood, softwood and crop residuals was introduced from work of Cheng et al. (2013), which overcomes the limitation of using only one characteristic ratio (either lev/nss-$K^+$ or lev/man) to distinguish types of biomass being burned and hence increases the reliability of determination. In previous studies, emissions from the crop residuals burning were characterized by a lower lev/nss-$K^+$ ratio (mostly less than 1) (Sheesley et al.,

2003; Sullivan et al., 2008; Engling et al., 2009; Oanh et al., 2011) and a higher lev/man ratio which was reported ~20 in general and could be as high as 41 (Sheesley et al., 2003; Sullivan et al., 2008; Engling et al., 2009; Oanh et al., 2011). In the present study, the mean lev/man ratio is $22.5 \pm 12.3$ and the mean lev/nss-$K^+$ is $0.3 \pm 1.2$. According to Fig. 6, ~94 % of the ambient samples (96 out of 102) in this study are traced to crop residuals & grass region, demonstrating that crop residuals were main biomass type burned in the investigated episode, since the contribution of grass was negligible compared

to the total biomass consumed in China (Streets et al., 2003). Our result is similar to that reported for the typical biomass burning events in summer in Beijing (Cheng et al., 2013).

### 3.5.2 Long-range transport

In order to examine the influence of long-range transport on local atmospheric compositions, 48 h backward trajectories of air masses reaching the Nanjing sites at a height of 500 m were computed via the HYSPLIT model covering all the sampling days.

Figure 7 displays the MODIS fire spot distribution during 21 Jan to 25 Jan, when open burning was strong in China. No fire signals are found in the local region of Nanjing, suggesting the dominant influence from long-range transport. Some fire spots appeared in northern China but densely distributed spots were in southeastern China. All 48 h backward air mass trajectories are classified into 4 clusters and mean levoglucosan concentrations of all trajectories for each cluster are described in Fig. 7. The cluster originating from southeast China had the highest mean levoglucosan concentration (796 ng m$^{-3}$) among the four

clusters, although it transported the least air mass to Nanjing (20 %). Therefore, it is very likely that the biomass burning contribution to aerosols in this study was attributed to the emission from southeast China.

Moreover, when the maximum levoglucosan was observed between 05:00 and 08:00 on 25 Jan, the Flexpart potential emission sensitivity in Fig. 8 indicates that the air masses were mainly coming from the south and southeast regions. It is evident that air masses caught the pollutants emitted from biomass burning on 21-24 Jan, as was illustrated by the hotspots.

Notably, there is a cluster originating from the coastal site of eastern China mainly passing through Shanghai and Jiangsu, where no fire signals are found. But this cluster has a relatively high mean levoglucosan concentration (673 ng m$^{-3}$). According to the research by Zhou et al. (2017) on biomass burning emission inventory, domestic straw burning was responsible for ~42




% of total biomass burning in China in 2012, and ~1.5 Gg and ~80 Gg PM$_{2.5}$ came from domestic burning in Shanghai and Jiangsu, respectively. It illustrated the significant impact of indoor biomass burning which was invisible in MODIS fire spots map. Therefore, the levoglucosan enrichment of the cluster from coastal area might be due to domestic biomass burning. Even if local domestic biomass burning proves to be an important contributor, the effects of long-distance transmission demonstrated above cannot be ignored.

### 3.6 Light-absorbing property

The light absorption coefficient of WSOC at 365nm is used to represent water-soluble BrC, which is organic carbon with a capacity of absorbing radiation (Yan et al., 2015; Hecobian et al., 2010; Wang et al., 2018a). The mean b$_{abs}$ value was 9.4 ± 4.8 M m$^{-1}$, higher than the summertime value but lower than the wintertime value in Beijing and Guangzhou (Yan et al., 2015; Qin et al., 2018). Meanwhile, it has a large span from 1.6 to 30.0 M m$^{-1}$, partially attributed to its dependence on biomass burning intensity. As shown in Fig. 9, b$_{abs}$ is significantly correlated with biomass burning tracers ($R^2 > 0.5$, $p < 0.01$), indicating that a considerable quantity of BrC was associated with biomass burning emission in this episode. According to the analysis in Sect. 3.4, the correlation implied that biomass burning has contributed to WSOC. With respect to this point, most WSOC released by biomass burning could be BrC, as suggested by the strong correlation between b$_{abs}$ and WSOC in Fig. 10 ($R^2 = 0.76$, $p < 0.01$). However, biomass burning cannot be the only origin of BrC, because the b$_{abs}$ is poorly correlated to the BB-WSOC ratio in Fig. 10 ($R^2 = 0.04$, $p < 0.05$). Other researchers pointed out that secondary photochemical reactions in the atmosphere can produce BrC as well (Sareen et al., 2013; Liu et al., 2016).

It is evident that the absorption coefficient of WSOC shows a relationship with wavelength, which can be described by absorption Angstrom exponent. AAE is also used to distinguish the BrC emission types due to its dependence on particle size and composition. Generally, the Angstrom exponent for compounds from biomass burning and biofuel emissions is larger than 6 (Hoffer et al., 2006; Chen and Bond, 2010; Hecobian et al., 2010; Bikkina, 2014). The average AAE in this study was 6.6 ± 1.3, ranging from 4.1 to 10.3, with more than 68% of samples (71 out of 104) having an AAE above 6 (Fig. 13). Our result is consistent with those for the Indo-Gangetic Plain, where biomass burning emission has been proved as the predominant aerosol source (Bikkina, 2014). If light absorption is dominated by elemental carbon, AAE is reported to be ~1 (Bond, 2001; Kirchstetter et al., 2004). For fresh and aged SOA it should be ~7 and ~4.7, respectively (Bones et al., 2010).

Mass absorption efficiency characterizes the efficiency of light absorption by WSOC. As shown in Fig. 11, the average MAE in this study was 1.0 ± 0.2 m$^2$ g$^{-1}$, with a range of 0.5-1.6, higher than that of some US cities, Beijing (summer) and the Indo-Gangetic Plain effected by biomass burning, but lower than that reported for some Indian areas and Beijing, China during wintertime. The MAE was lower than 1.79 m$^2$ g$^{-1}$ in Beijing during winter. As shown in Fig. 11, MAE has poor correlation with AAE (Yan et al., 2015; Kirillova et al., 2014b; Kim et al., 2016; Du et al., 2014b; Hecobian et al., 2010; Yan et al., 2017; Cheng et al., 2011; Zhang et al., 2013; Bikkina, 2014; Srinivas et al., 2016; Cheng et al., 2016).

As shown in Fig. 13, solar energy absorption of WSOC (E$_{WSOC}$) in short wavelengths (300-400 nm) had a mean value of 0.8 ± 0.4 W m$^{-2}$, ranging from 0.2 to 2.3 W m$^{-2}$. The counterpart contributed by biomass burning (E$_{WSOC\_BB}$), derived from multiplying E$_{WSOC}$ by BB-WSOC, had a mean value of 0.2 ± 0.1 W m$^{-2}$, ranging from 0.0 to 0.9 W m$^{-2}$. Solar energy absorption



of EC ($E_{EC}$) in the wavelength range of 300-400 nm had a mean value of $3.4 \pm 1.7$ W m$^{-2}$, ranging from 0.4 to 9.9 W m$^{-2}$. The WSOC to EC absorption percentage ($E_{WSOC}/E_{EC}$) was $23.8 \pm 8.7\%$ and could be up to 73.0% at the short wavelengths (300-400 nm), which is much higher than that for Beijing winter (42%) (Yan et al., 2015). The map in Fig. 12 specifies the level of PSCF values related to the carbonaceous compounds (levoglucosan, WSOC and EC) and their radiative absorption properties

($b_{abs}$, $E_{WSOC}$, $E_{EC}$, $E_{WSOC}/E_{EC}$) in terms of a colour bar. The areas with high PSCF values were interpreted as the potential areas loading the carbonaceous compounds or better light absorption. As shown in Fig.12, the water-soluble carbonaceous aerosols transported to Nanjing over a long distance had a stronger light absorption capacity, compared to those aerosols from local emissions. It is evident that WSOC would enhance its light absorption capacity after aging along the long-range air mass transport and the enhancement is much more than that for EC (Kirillova et al., 2014a). It means that the carbonaceous emissions

over long-range transport can make more contribution to affect the local atmospheric solar radiation balance. In addition, the levoglucosan has a similar PSCF distribution with WSOC and EC, illustrating that long-range transported biomass burning emission profoundly impacted the carbonaceous aerosols in Nanjing during this episode. The similar PSCF distributions between WSOC and $b_{abs}$ or $E_{WSOC}$ can be explained by the fact that light-absorption of WSOC is usually related to WSOC levels. The similarity of PSCF distributions between EC and $E_{EC}$ can be explained in the same way. However, the PSCF

distribution of $E_{WSOC}/E_{EC}$ is different from $E_{EC}$ or $E_{WSOC}$ in Fig. 12, indicating southeastern China was the potential areas loading the aerosols with a higher light-absorbing contribution by WSOC. It is consistent with the fire spot distribution in Fig. 7, which implies that the long-range transported biomass burning emission significantly impacted the solar energy absorption of carbonaceous aerosols in Nanjing during this episode.

In Table 3, we show the optical properties (MAE*, SSA* and g*) for EC and WSOC (at RH=0.7) in the band 26 (22650-29000

20 cm$^{-1}$, i.e., 345-442 nm) as provided in RRTMG_SW model. Based on the surface mass concentration of EC and WSOC measured in this study, RRTMG_SW simulated the respective absorptive properties in the first layer of the model (the lowest layer above the ground), which accounts for the aerosol absorption in the radiative transfer calculation. To obtain the absorptive radiative forcing (ARF) of EC or WSOC, RRTMG_SW was run twice by including and excluding EC or WSOC in the model. The differences in the radiative flux between the two runs indicate the adsorptive radiative effect of EC or WSOC in the

25 atmosphere.

In the wavelength range of 345-442 nm, the change in the net flux at the surface caused by absorption of EC and WSOC was $-3.4 \pm 1.6$ and $-1.4 \pm 0.6$ W m$^{-2}$ respectively, which are consistent to the previously estimated $E_{EC}$ and $E_{WSOC}$. The ARF at the top of the atmosphere (TOA) was $0.8 \pm 0.4$ W m$^{-2}$ for EC and $-0.2 \pm 0.1$ W m$^{-2}$ for WSOC. Therefore, the ARF in the atmosphere (ATM) caused by absorption of EC ($E^*_{EC}$) and WSOC ($E^*_{WSOC}$) was $4.2 \pm 2.0$ W m$^{-2}$ and $1.2 \pm 0.5$ W m$^{-2}$,

respectively, which are slightly higher than the previously estimated $E_{EC}$ and $E_{WSOC}$ indicating that the absorbing aerosols maintain a part of the solar energy in the atmosphere. Our estimates from model indicate that the absorption ability of WSOC was equivalent to about 30% of EC absorption in this study, which is comparable to the ratio from previous studies. For example, Lin et al. (2014) simulated the global OC radiative forcing and indicated that the absorption due to OC (both primary and secondary OC) aerosols is about 27-70% of EC warming effect in the global atmosphere. Liu et al. (2014b; 2015)

investigated the absorption of OC aerosols at different altitudes over the North American continent and found that OC



absorption at TOA is around 20% of EC direct effect in the background troposphere. Our study provides an additional evidence and stresses the necessity to account for the absorption capacity of WSOC in model simulations of the global energy budget.

## 4. Conclusions

In this study, biomass burning tracers (including levoglucosan, mannosan, galactosan and nss-$K^+$) were used to quantify biomass burning contributions to the carbonaceous components of $PM_{2.5}$ in Nanjing during a wintertime pollution event. The origin of biomass burning and biomass fuel types were also investigated. Furthermore, solar energy absorption due to biomass-burning BrC was quantified.

The levoglucosan concentration in $PM_{2.5}$ in this study was up to 1476 ng m$^{-3}$, which is extremely higher in comparison with those found in other studies. Biomass burning contribution to OC and WSOC was, on average, $20.9 \pm 9.3$ % and $22.3 \pm 9.9$ %, respectively, and was as high as 53.6 % to OC and 55.4 % to WSOC. Both of the facts indicate the significant impact of biomass burning in this atmospheric pollution episode, although no local open biomass burning was found during the study period. The combination of long-range transport analysis, fire spots information and the potential emission sensitivity of BC suggests that the burning of crop residuals in southeast China was responsible for the significant biomass combustion to the ambient aerosols.

The mass absorption efficiency of water-soluble BrC at the 365 nm wavelength was averaged $1.0 \pm 0.2$ m$^2$ g$^{-1}$, with a range from 0.5 to 1.6 m$^2$ g$^{-1}$. Based on the measured light-absorbing properties, solar energy absorption of WSOC and EC in short wavelengths (300-400 nm) was calculated to be $0.8 \pm 0.4$ W m$^{-2}$ and $3.4 \pm 1.7$ W m$^{-2}$, respectively, which were slightly lower than $1.2 \pm 0.5$ W m$^{-2}$ for WSOC and $4.2 \pm 2.0$ W m$^{-2}$ for EC, the results simulated by a radiative transfer model (RRTMG_SW). The maximum solar energy absorption of WSOC due to biomass burning was 0.9 W m$^{-2}$. It provides evidence that carbonaceous components released by biomass burning transported over long distance absorbed solar energy, especially in the UV band, influencing the radiation balance of the atmosphere. The PSCF analysis of carbonaceous compounds concentration and light-absorption of carbonaceous aerosols show that the regional transported biomass burning emissions profoundly impacted the chemical and optical properties of carbonaceous aerosols in Nanjing during this episode.

*Author contributions.* XL and YZ conceived the study. XL wrote the manuscript with YP, LX and CZ. XL, XZ, TH and ZX carried out the experiments and collected data. All have contributed to the data interpretation and review of the manuscript.

*Competing interests.* The authors declare that they have no conflict of interest.

*Acknowledgements.* This study was financially supported by the National Key R&D Program of China (Grant No. 2017YFC0210101), the Natural Scientific Foundation of China (No. 91643109 and 41603104), the Provincial Natural Science Foundation of Jiangsu (Grant No. BK20180040) and China Scholarship Council.



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





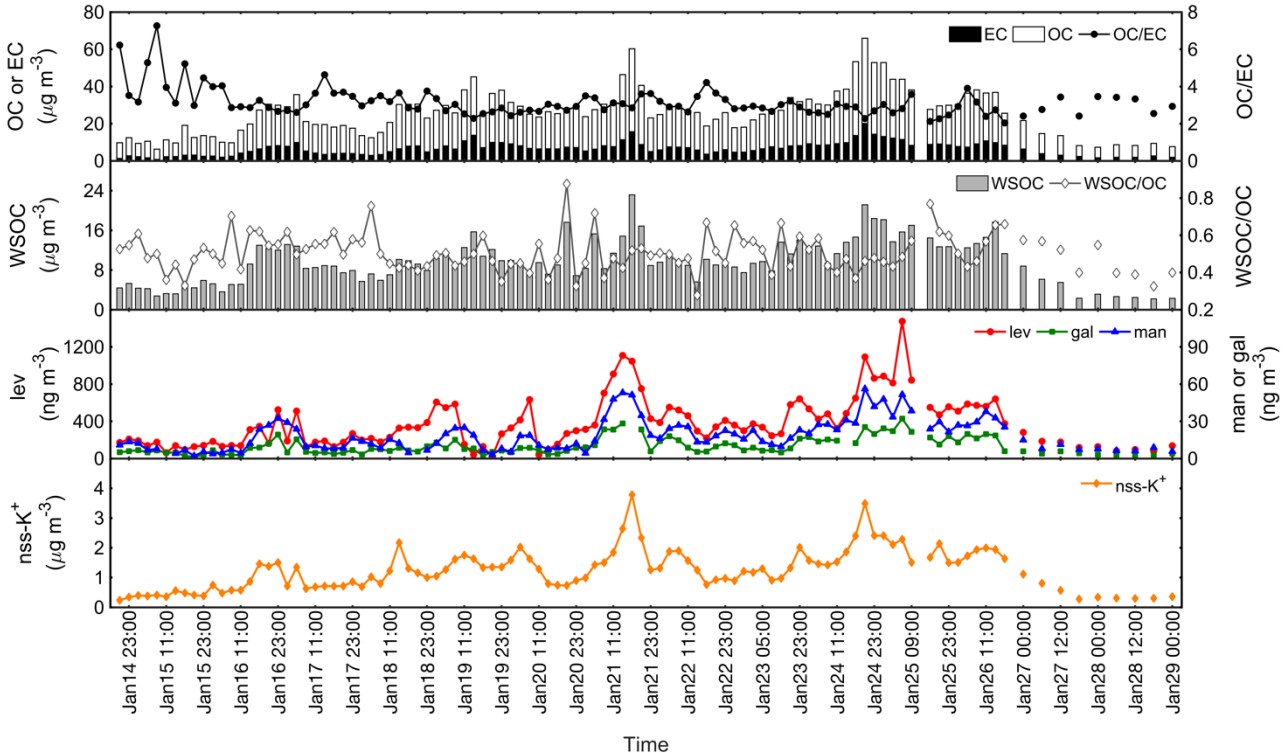

**Figure 1: Time series of OC, EC, WSOC, levoglucosan (lev), mannosan (man), galactosan (gal) and non-sea-salt potassium (nss-K$^+$) concentration and OC/EC, WSOC/OC ratio for PM$_{2.5}$ samples. The horizontal ordinate indicates ending time for each sample.**




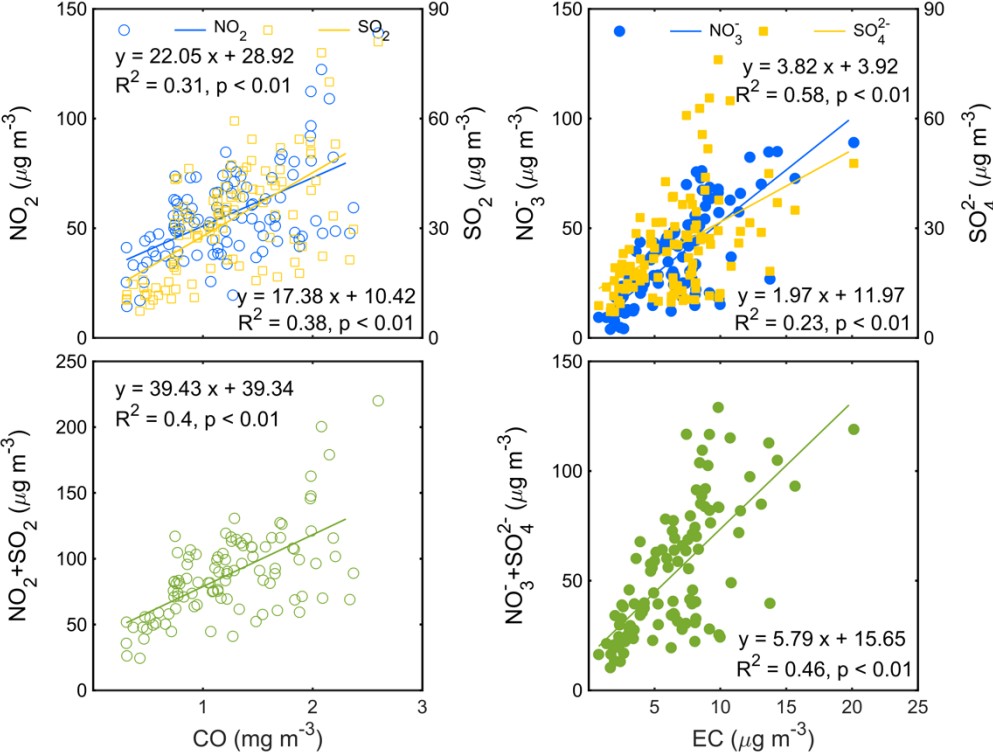

**Figure 2: Correlations of carbon monoxide (CO) with nitrogen dioxide (NO₂), sulfur dioxide (SO₂) and sum of nitrogen dioxide and sulfate dioxide (NO₂ + SO₂) in ambient air, and correlations of EC with nitrate (NO₃⁻), sulfate (SO₄²⁻) and sum of nitrate and sulfate (NO₃⁻ + SO₄²⁻) in PM₂.₅ from 14 to 29 Jan, 2015 in Nanjing.**





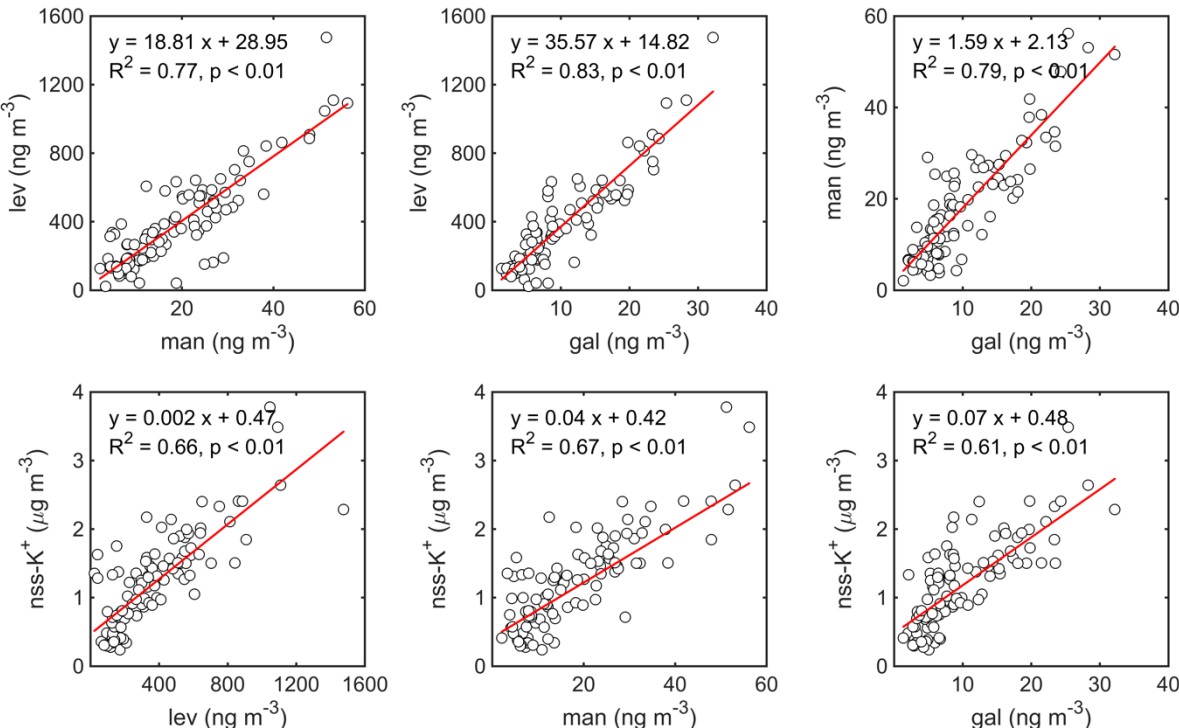

**Figure 3: Correlations between 4 tracers of biomass burning including levoglucosan (lev), mannosan (man), galactosan (gal) and non-sea-salt potassium (nss-K$^+$) in PM$_{2.5}$ from 14 to 29 Jan, 2015 in Nanjing.**





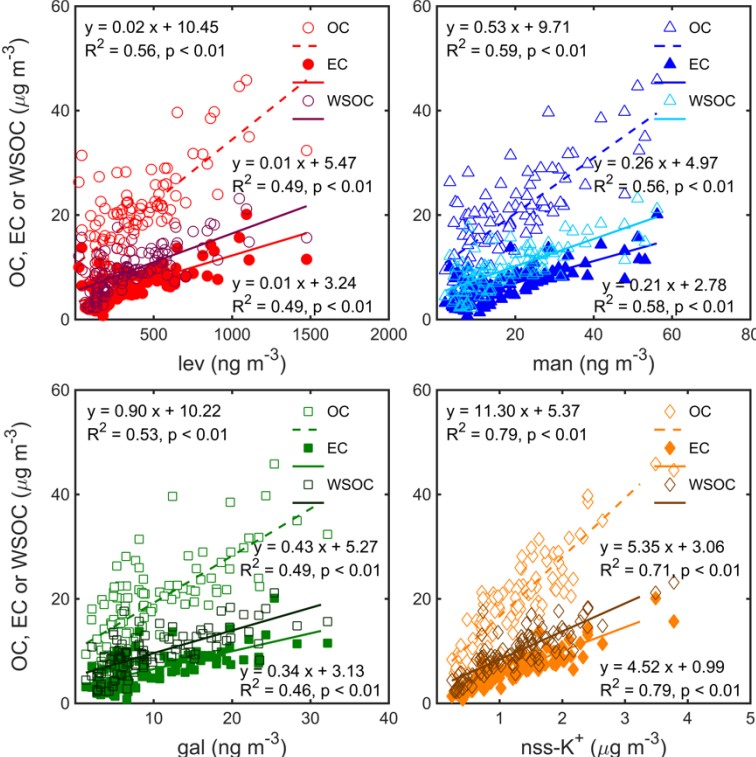

**Figure 4: Correlations of OC, EC and WSOC vs levoglucosan (lev), mannosan (man), galactosan (gal) and non-sea-salt potassium (nss-K⁺) in PM$_{2.5}$ from 14 to 29 Jan, 2015 in Nanjing.**




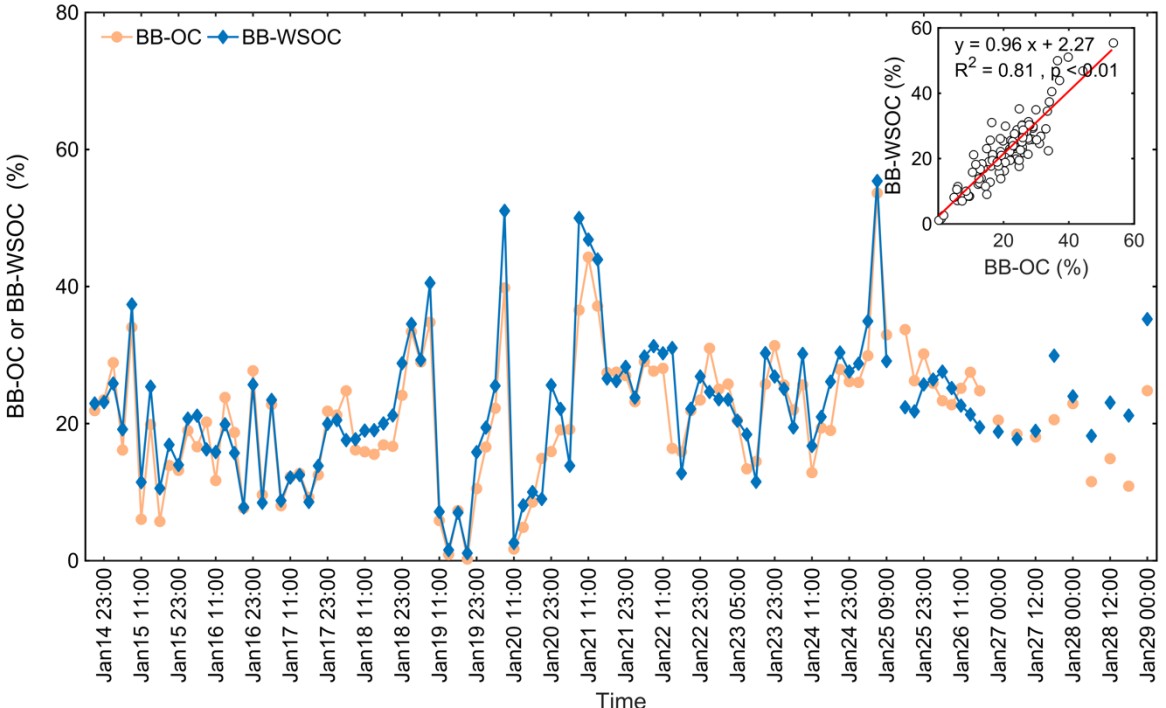

**Figure 5: Time series of biomass burning contribution to OC (BB-OC) and WSOC(BB-WSOC).**



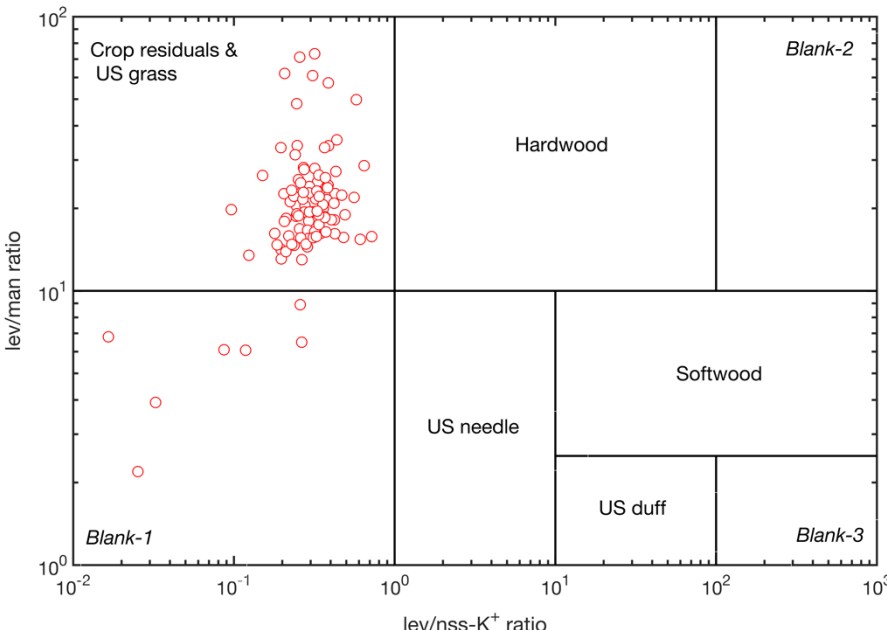

**Figure 6: Representative ranges of lev/nss-K+ and lev/man ratios for different types of biomass fuel introduced by Cheng et al. (2013). Results from the ambient samples collected in this study are also shown for comparison.**





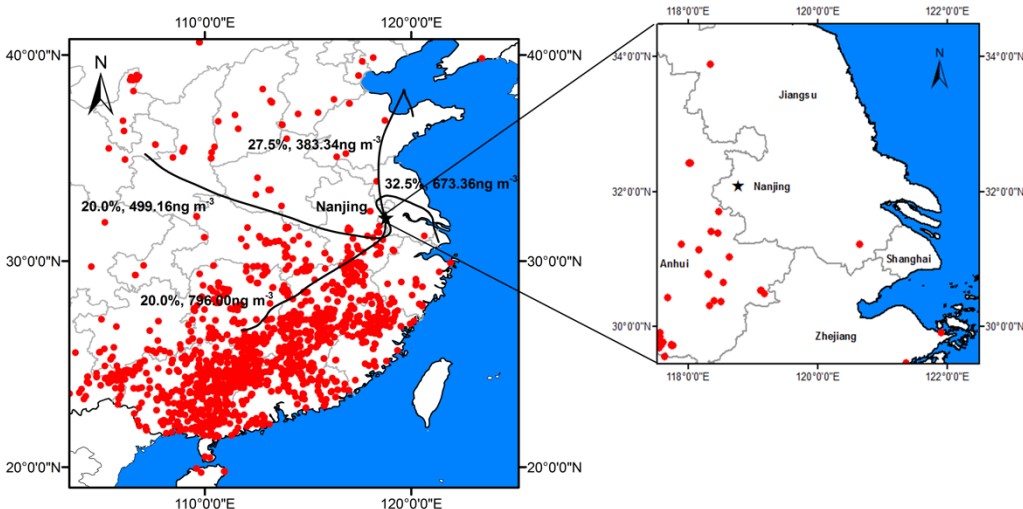

**Figure 7: Air mass backward trajectory from 21 to 25 Jan, along with the MODIS fire spot map. Percentage refers to the air mass contribution to Nanjing and values with a unit of ng m$^{-3}$ refers to mean concentration of levoglucosan for every trajectory in that cluster.**



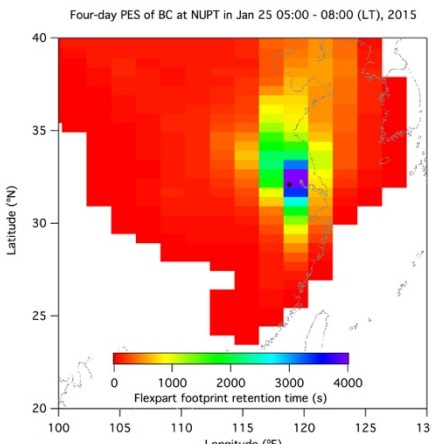

**Figure 8: The four-day footprint of NUPT (118.78° E, 32.09° N, black point), the observation site, starting from 25 Jan, 05:00-08:00 LT when the maximum levoglucosan was observed. The footprint was illustrated as Flexpart potential emission sensitivity shown as the retention time in each grid at 0–500 m contributing to the observed elemental carbon.**





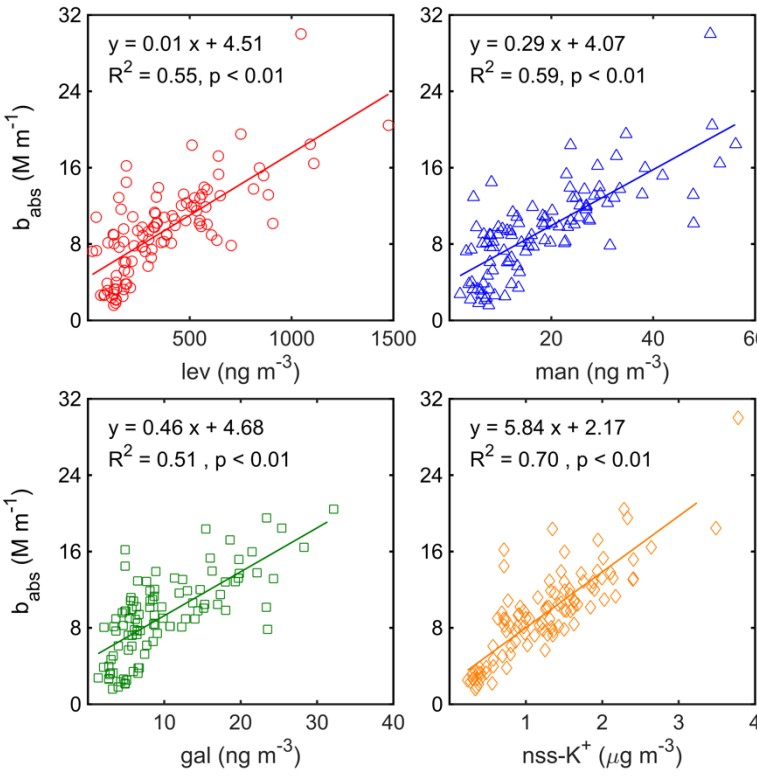

**Figure 9: Correlations of absorption coefficient ($b_{abs}$) vs levoglucosan (lev), mannosan (man), galactosan (gal) and non-sea-salt potassium (nss-K$^+$) in PM$_{2.5}$ from 14 to 29 Jan, 2015 in Nanjing.**





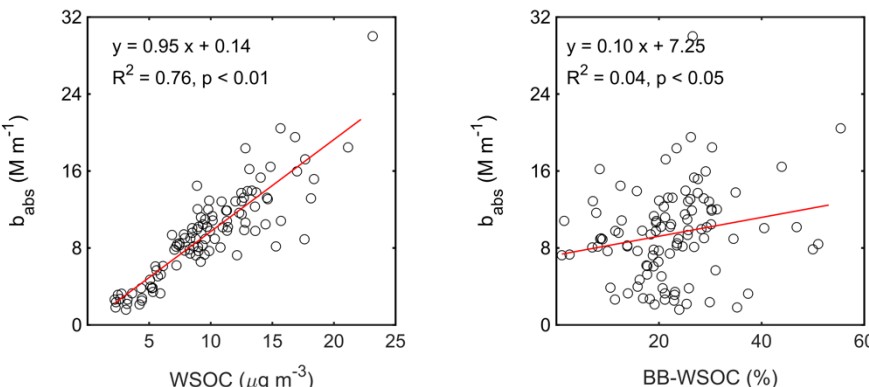

**Figure 10: Correlations of WSOC and biomass burning contribution to WSOC (BB-WSOC) vs absorption coefficient ($b_{abs}$) in PM$_{2.5}$ from 14 to 29 Jan, 2015 in Nanjing.**



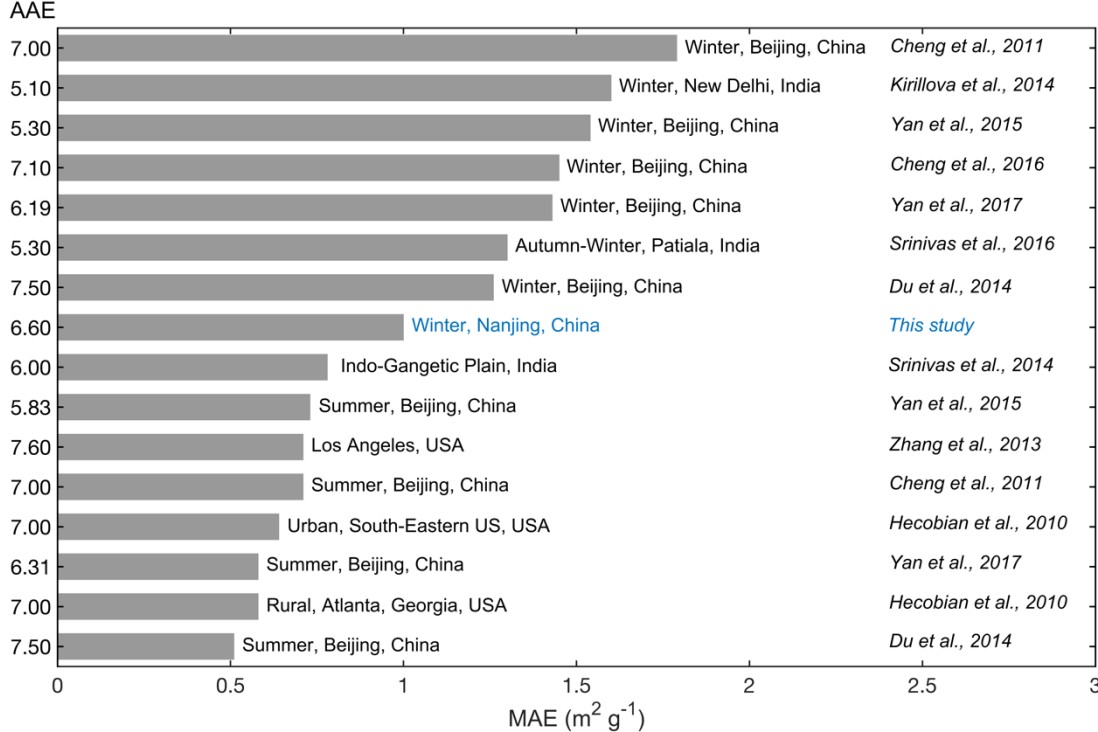

**Figure 11: Comparison of mean mass absorbance efficiency (MAE) and absorption Angstrom exponent (AAE) of WSOC in PM$_{2.5}$ in this study with those reported in the literature.**

**Figure 12: The PSCF map for levoglucosan (lev), absorption coefficient (b$_{abs}$), WSOC, EC, light-absorption of WSOC (E$_{WSOC}$), and EC (E$_{EC}$), WSOC to EC light-absorption ratio (E$_{WSOC}$/E$_{EC}$) for PM$_{2.5}$ from 14 to 29 Jan, 2015 in Nanjing (black point).**





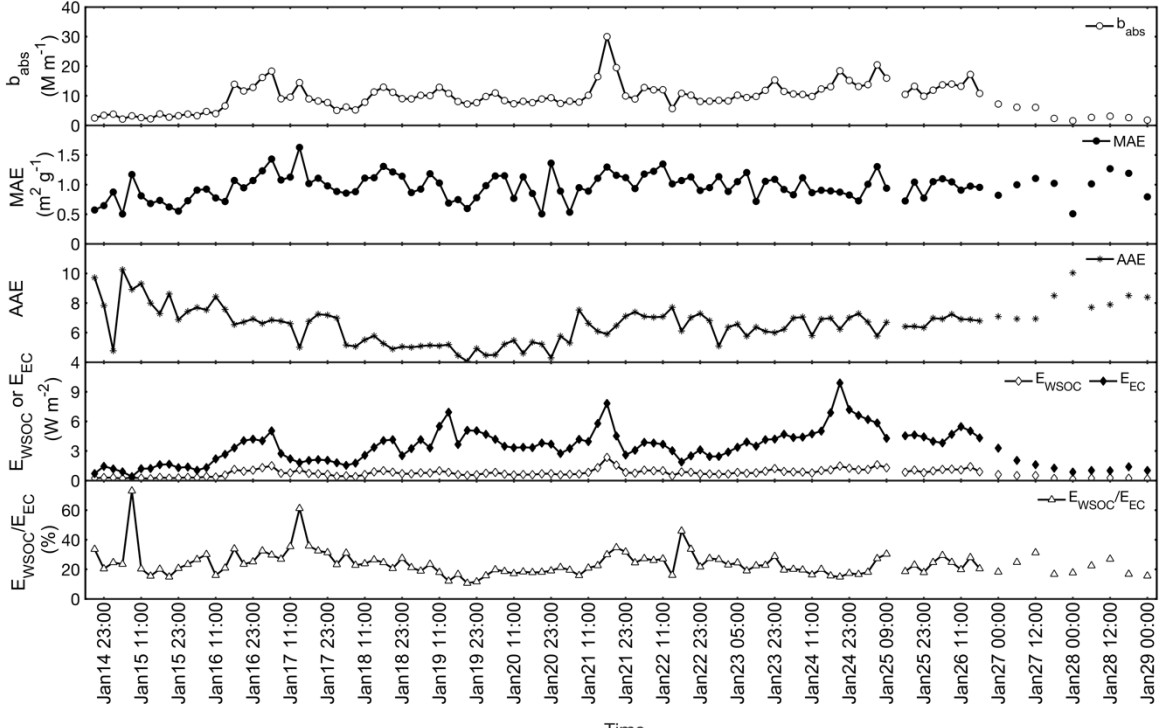

**Figure 13: Time series of absorption coefficient ($b_{abs}$), mass absorbance efficiency (MAE), absorption Angstrom exponent (AAE), light-absorption of WSOC ($E_{WSOC}$) and EC ($E_{EC}$), WSOC to EC light-absorption ratio ($E_{WSOC}/E_{EC}$) for PM$_{2.5}$ from 14 to 29 Jan, 2015 in Nanjing.**



**Table 1. Concentration and mass ratio of atmospheric gaseous components and particulate species (PM$_{2.5}$), biomass burning contribution to carbonaceous aerosol, light-absorption parameters of water-soluble BrC and solar energy absorption of carbonaceous aerosols.**

|  |  | MEAN | STD | MIN | MAX |
|---|---|---|---|---|---|
| CO | mg m$^{-3}$ | 1.2 | 0.5 | 0.3 | 2.6 |
| NO2 | µg m$^{-3}$ | 56.1 | 21.0 | 14.3 | 139.0 |
| SO2 | µg m$^{-3}$ | 31.8 | 15.1 | 7.3 | 81.0 |
| OC | µg m$^{-3}$ | 19.4 | 8.6 | 5.5 | 45.8 |
| EC | µg m$^{-3}$ | 6.6 | 3.5 | 0.8 | 20.1 |
| WSOC | µg m$^{-3}$ | 9.7 | 4.3 | 2.2 | 23.2 |
| OC/EC |  | 3.2 | 0.8 | 2.0 | 7.3 |
| WSOC/OC |  | 0.5 | 0.1 | 0.3 | 0.9 |
| NO$_3^-$ | µg m$^{-3}$ | 29.3 | 17.4 | 3.1 | 71.2 |
| SO$_4^{2-}$ | µg m$^{-3}$ | 24.8 | 14.1 | 7.1 | 76.1 |
| nss-K$^+$ | µg m$^{-3}$ | 1.2 | 0.7 | 0.2 | 3.8 |
| levoglucosan | ng m$^{-3}$ | 373 | 268 | 22.4 | 1476 |
| mannanson | ng m$^{-3}$ | 18.5 | 12.5 | 2.1 | 56.2 |
| galactosan | ng m$^{-3}$ | 9.9 | 6.7 | 1.4 | 32.2 |
| BB-OC | % | 20.9 | 9.3 | 0.2 | 53.6 |
| BB-WSOC | % | 22.3 | 9.9 | 1.1 | 55.4 |
| lev/man |  | 22.5 | 12.3 | 2.2 | 73.2 |
| lev/nss-K$^+$ |  | 0.3 | 0.1 | 0.0 | 0.7 |
| b$_{abs}$ | M m$^{-1}$ | 9.4 | 4.8 | 1.6 | 30.0 |
| MAE | m$^2$ g$^{-1}$ | 1.0 | 0.2 | 0.5 | 1.6 |
| AAE |  | 6.6 | 1.3 | 4.1 | 10.3 |
| E$_{WSOC}$ | W m$^{-2}$ | 0.8 | 0.4 | 0.2 | 2.3 |
| E$_{EC}$ | W m$^{-2}$ | 3.4 | 1.7 | 0.4 | 9.9 |
| E$_{WSOC\_BB}$ | W m$^{-2}$ | 0.2 | 0.1 | 0.0 | 0.9 |
| E$_{WSOC}$/E$_{EC}$ | % | 23.8 | 8.7 | 10.6 | 73.0 |
| E$^*_{WSOC}$ | W m$^{-2}$ | 1.2 | 0.5 | 0.3 | 1.9 |
| E$^*_{EC}$ | W m$^{-2}$ | 4.2 | 2.0 | 1.5 | 7.8 |

$^*$ refers to values derived from RRTMG_SW model simulation.



**Table 2. Comparison of levoglucosan concentration range in PM$_{2.5}$ during this study with those reported in the literature.**

| Sampling Site | Site Type | Sampling Time | Concentration Range (ng m$^{-3}$) | Reference |
|---|---|---|---|---|
| Azores, Portugal | coastal | Winter | 6.6 | |
| Puy de Dôme, France | coastal | Winter | 18.3 | (Puxbaum et al., 2007) |
| Schauinsland, Germany | coastal | Winter | 33.7 | |
| Sonnblick, Austria | coastal | Winter | 12.4 | |
| Jilin, China | forest | July | 42 (32-67) | |
| Shanghai, China | forest | June | 143 (20-212) | (Wang et al., 2008) |
| Guangdong, China | forest | August | 25 (0.3-61) | |
| Hainan, China | forest | November | 107 (19-398) | |
| Beijing, China | urban | Summer | 230 | |
| | | BB episode | 750 | |
| | | Typical summer | 120 | (Cheng et al., 2013) |
| | | Winter | 590 | |
| | | Firework episode | 460 | |
| | | Typical winter | 640 | |
| Shanghai, China | urban | Spring | 66 (18-159) | |
| | | Summer | 28 (8.6-194) | |
| | | Autumn | 229 (13-1606) | (Li et al., 2016c) |
| | | Winter | 161 (26-614) | |
| Nanjing, China | urban | Winter | 373 (22.4-1476) | Current study |



**Table 3. Optics for EC and WSOC (RH=0.7) in the shortwave band 26 (22650-29000 cm$^{-1}$, 345-442 nm) provided in RRTMG_SW.**

|  | MAE$^*$ (m$^2$ g$^{-1}$) | SSA$^*$ | g$^*$ |
| --- | --- | --- | --- |
| EC | 10.75 | 0.27 | 0.40 |
| WSOC (RH=0.7) | 2.44 | 0.66 | 0.71 |

