# Peer review of "Chemical and Optical Properties of Carbonaceous Aerosols in Nanjing, East China: Regional-transported Biomass Burning Contribution"

_Atmospheric Chemistry and Physics, 2018_

## Referee Comment (RC1) · Anonymous Referee #1 · 27 Mar 2019

This study investigates the chemical and optical properties of carbonaceous aerosols in Nanjing and finds that the transport of biomass burning emissions is a major source of carbonaceous aerosols. The paper is clear on describing the results they found, and the conclusion is solid. My largest concern is the way how the authors present the paper. From my understanding, the biomass burning emissions are transported to atmosphere in Nanjing and they showed different properties than other episodes, it is not the case that the transported emission changed the properties of aerosols emitted in Nanjing. Thus, I would avoid saying that the "Impacts of Regional-transported Biomass

Burning Emissions on", which is misleading. Detailed comments are below.

Comments:

1. Page 1: Abstract: "Biomass burning can significantly impact the chemical and optical properties of carbonaceous aerosol", this sentence is repeating with the next one.

2. Page 2, Lines 5-6, please avoid repeating such blurry information.

3. Page 2, Lines 21: long-range transport of aerosols; "simulation modeling"?

4. Page 2, Lines 26: methods for biomass burning calculation

5. Page 3, Line 20, POM?

6. Page 3, Line 21, This is no turning here to use "However"

7. Page 4, Line 2, why WSOC contribute to EC?

8. Page 4, Line 4, what are MAC365 and AAE

9. Page 10, the equations to estimate the contributions of biomass burning to carbonaceous components are based on the emissions of biomass burning. It should be discussed about the uncertainties as the ambient concentrations measured in Nanjing are mixture of biomass burning transported to here and local emissions from other sources.

10. Page 11, the levoglucosan contration could be as high as 673 ngm-3 from non-biomass burning regions, you cannot put all 796 ngm-3 from southeast China to biomass burning.

11. Figure 10: from the relationship of OC, EC, and WSOC with lev, man, gal and nss-K in Figure 4, WSOC is correlated with biomass burning. Then, it is not clear why babs is not correlated with BB-WSOC at all (Figure 10), and it can not be explained by biomass burning is not the only source of BrC.

12. Contributions of biomass burning to carbonadoes aerosols are a key factor for analyzing their radiative effects. The authors should add these key information in abstract or conclusion.

13. Figure 8 is not in high quality, where is Nanjing? I cann't agree with the authors that it shows "the air masses were mainly coming from the south and southeast regions".

14. Table 2: I can understand that the authors want to compare with other studies, but I don't believe there are only four previous studies have reported lev concentration in the whole literature.
* * *

---

## Referee Comment (RC2) · Anonymous Referee #2 · 29 Mar 2019

The study quantitatively evaluated the influence of long-range transport of biomass burning (BB) emissions on carbonaceous aerosols in Nanjing city, China, which is helpful to understand the sources and associated contributions to aerosol carbonaceous component in urban regions in China. Following issues need to be addressed prior to publication.

1. One concern is about the relative contributions of long-range transport BB versus the local domestic BB emissions shown in Fig. 7. Though the levoglucosan concentration contained in air masses from southeast China is higher than that from the coastal

region of eastern China, the air mass frequency from the latter air mass cluster (32.5%) is much higher than the former one (20%). After weighted with the cluster frequency, I would say the contribution from coastal eastern China actually is larger than that from southeast China. I understand that the long-range BB could contribute to aerosols in Nanjing significantly during certain time period, but, from long-term perspective, it might not be the case due to the large contribution from local domestic BB emissions. As such, the authors may want to make changes on the statements related to the contribution of long-range contribution.

2. The physical and optical properties of black carbon after emitted into atmosphere for certain time can be largely modified by the aging process (e.g., Peng et al., PNAS, 2016; Wang et al., JAMES, 2018). Please add some reviews regarding soot aging in the introduction section on page 4 when reviewing the alterations of physical and optical properties with the transport process.

3. In abstract: when talking about absorption of carbonaceous aerosols, the two terms (i.e., WSOC and water soluble BrC) are basically equivalent, but the authors switched back and forth to use them. Please stick to one expression.

4. Since the observations are ground-based, while the back-trajectory analysis focuses on 500m level. Are the conclusions about relative contributions from different air mass clusters sensitive to the selected altitude?

5. Typos Line 26 page 2: missing a preposition at ". . .frequently method biomass burning. . ."

---

## Referee Comment (RC3) · Anonymous Referee #3 · 4 Apr 2019

The manuscript measured the chemical and optical properties of carbonaceous aerosols in Nanjing and attempted to elaborate impacts of the biomass burning (BB) originated from regional transport on the carbonaceous aerosols. The topic is of interest but the manuscript is not well written. There are several factors hinder publication of the manuscript at the present form, and a major revision is needed.

Major comments:

1. The title is "Impact of regional-transported biomass burning emissions on chemical

and optical properties of carbonaceous aerosols in Nanjing, East China". Therefore, It is assumed that the measured aerosol properties in Nanjing with and without BB impact should have large difference. Figure 5 shows that there are moments with low BB impact on Nanjing. Did the chemical and optical properties of carbonaceous aerosols under the low BB influence reveal a significant difference from those under the high BB influence? 2. Figure 5 shows that the fraction of BB-OC and BB-WSOC is nearly the same throughout the measurement. Does this mean that the emission ratios of WSOC/OC from other anthropogenic sources are the same as BB, or is it the coincidence from the calculation by the equation 6 and 7? 3. Since the authors attempted to investigate the impact of BB on carbonaceous aerosols, why did the authors only measure the absorption of BrC in WSOC, instead of the absorption of total BrC in OC? There are also water-insoluble BrC from BB. 4. Page 8, Line 17, the authors concluded that secondary OC is weak and stable only based on the OC/EC ratio ranging from 2 to 4 during the severe pollution period. The conclusion is arbitrary. Firstly, the OC/EC ratio ranging from 2 to 4 are rather large. Secondly, previous studies have indicated that SOA contribution during haze days during wintertime in China is considerable. For example, Huang et al. (2014) showed SOA contributed 44%-71% to the OA mass in four megacities of China during wintertime. Sun et al. (2013) revealed that SOA contributed 31% to the OA mass in Beijing in winter.

Minor comments:

Page 2, Lines 11-15: What is the difference among those portions of OC? Page 2, Lines 15-16: Does it indicate that BrC is more absorptive than BC? Page 2, Line 26: what is "the most frequently method"? Page 3, Line 3, "that" should be removed? Page 3, Line 18, "impact" should read "impacts". Page 3, Line 21: "than" should read "compared to". Page 3, Lines 25-27: Are the three parameters are specially used for BrC from BB? Page 3, Lines 28: What is HULIS? Page 7, Line5, "give" should be "given". Page 8, Line 10, "uncertainty" should be "variation" Page 10, Line 25, "which illustrating the necessity of parameter BB-OC". The sentence is hard to understand.

References: Huang et al., High secondary aerosol contribution to particulate pollution during haze events in China, Nature, 514, 218–222, https://doi.org/10.1038/nature13774, 2014. Sun et al., Aerosol composition, sources and processes during wintertime in Beijing, China, Atmos. Chem. Phys., 13, 4577–4592, https://doi.org/10.5194/acp-13-4577-2013, 2013.

---

## Author Comment (AC1) · 16 May 2019

RE: A point-to-point response to reviewers' comments

'Impacts of Regional-transported Biomass Burning Emissions on Chemical and Optical Properties of Carbonaceous Aerosols in Nanjing, East China' (acp-2018-1029) by Xiaoyan Liu, Yan-Lin Zhang, Yiran Peng, Lulu Xu, Chunmao Zhu, Xiaoyao Zhai, Chi Yang, Mozammel Haque, Fang Cao, Yunhua Chang, Tong Huang, Zufei Xu, Mengying Bao, Wenqi Zhang, Meiyi Fan, Xuhui Lee

We are grateful to the helpful comments by the referees, and have carefully revised our manuscript accordingly. A point-to-point response to the reviewers' comments is given below, in which comments are repeated in italic; the citations from the revised manuscript are underlined; the revised or added statements are highlighted in yellow. The revised manuscript copy is attached after the point-to-point response, in which the revised statements are highlighted in yellow.

Anonymous Referee #1

*This study investigates the chemical and optical properties of carbonaceous aerosols in Nanjing and finds that the transport of biomass burning emissions is a major source of carbonaceous aerosols. The paper is clear on describing the results they found, and the conclusion is solid. My largest concern is the way how the authors present the paper. From my understanding, the biomass burning emissions are transported to atmosphere in Nanjing and they showed different properties than other episodes, it is not the case that the transported emission changed the properties of aerosols emitted in Nanjing. Thus, I would avoid saying that the "Impacts of Regional-transported Biomass Burning Emissions on", which is misleading.*

Reply: We are grateful for the careful summery and kindly comments from the referee. According to the suggestion about the title, here's our reply:

In order to avoid being misleading, we would like to revise the title as 'Chemical and Optical Properties of Carbonaceous Aerosols in Nanjing, East China: Regional-transported Biomass Burning Contribution'. We agree with the reviewer in the sense that aerosols collected in situ observations are affected by both local emissions and from regional transported emissions such as during biomass burning episodes. This

study dedicated to figure out the role of regional transported biomass burning emissions in the pollution episode and it turned out that the chemical and optical properties of ambient aerosols in Nanjing was influenced by regional-transported biomass burning emissions during the episode because of the mixing of aerosols with diverse origins. In such a sense, we put 'aerosols in Nanjing' which directs general ambient aerosols in Nanjing atmosphere, instead of 'aerosols from local emissions of Nanjing'.

Detailed points:

1. *Page 1: Abstract: "Biomass burning can significantly impact the chemical and optical properties of carbonaceous aerosol", this sentence is repeating with the next one.*

Reply: As suggested, we removed the duplicated expressions and revised the description in Abstract as '==Biomass burning can significantly impact the chemical and optical properties of carbonaceous aerosols. Here, the biomass burning impacts were studied during wintertime in a megacity of Nanjing, East China.==' (Page 1 line 20-21).
The original sentences were 'Biomass burning can significantly impact the chemical and optical properties of carbonaceous aerosols. Here, the impacts of biomass burning emissions on chemical and optical properties of carbonaceous aerosols were studied during wintertime in a megacity of Nanjing, East China.'

2. *Page 2, Lines 5-6, please avoid repeating such blurry information.*

Reply: As suggested, we deleted the original sentence 'The regional transported biomass burning emissions may significantly impact the chemical and optical properties of carbonaceous aerosols in the polluted atmosphere', which indefinitely attributed the impacts of aerosols chemical and optical property to transported biomass burning emissions.

3. *Page 2, Lines 21: long-range transport of aerosols; "simulation modeling"?*

Reply: As stated in the original manuscript, the impacts due to biomass burning are regional and global owing to the aerosols long-range transport as 'All effects mentioned above are both regional and global owing to the aerosols long-range transport (Page 2 line 21-22)'. In order to figure out the role long-range

transport playing in the aerosol pollution events, numerical simulations were applied in previous studies, such as the references mentioned in our revised manuscript that '==Studies based on numerical simulations have shown that emissions from biomass burning can be transported to remote sites, even across the oceans (Aouizerats et al., 2015; Zhu et al., 2016; Ancellet et al., 2016)== (Page 2 line 22-23)'. Additionally, a clear statement including more details about numerical simulations research cases were added in our revised manuscript as '==Aouizerats et al. (2015) modeled a large fire-induced haze episode in 2006 stemming mostly from Indonesia using the Weather Research and Forecasting model coupled with chemistry (WRF-Chem) and found out the notable impact of long-range transported aerosols from Indonesia on ambient air quality and climate in Singapore. Ancellet et al. (2016) made the assessment of aerosol transport from North America to Europe using forward simulations of the FLEXPART Lagrangian particle dispersion model.==' (Page 2 line 23-27).

4. *Page 2, Lines 26: methods for biomass burning calculation*

Reply: The expression mistake has been corrected as suggested. The revised statements are 'In the observation-based studies, the characteristic ratio calculation and positive matrix factorization (PMF) are the most frequently methods ==to calculate== biomass burning ==contribution==. (Page 2 line 29-31)'.

5. *Page 3, Line 20, POM?*

Reply: 'POM' has been replaced of its full name '==primary organic matter==' (Page 3 line 24-25).

6. *Page 3, Line 21, This is no turning here to use "However"*

Reply: As suggested, 'However' has been replaced by 'Meanwhile' which better indicates the similar climatic effects between BC and BrC. helps to switch the emphasis from black carbon to brown carbon to further discuss the significant impact of brown carbon on radiative process. This paragraph is discussing the biomass burning aerosols impact on radiative forcing. Before the 'Meanwhile', it is describing the radiative forcing enhancement by black carbon, as 'Biomass burning aerosol has significant impacts on regional and global radiative forcing (Ramanathan and Carmichael, 2008). There is no doubt that element carbon enhances solar absorption of ambient aerosols most effectively (Ramanathan and Carmichael, 2008;

Myhre and Samset, 2015; Li et al., 2016b). Lack et al. (2012) found that the absorptivity of BC and primary organic matter mixture has 70% enhancement in biomass burning period compared to normal days in Boulder, Colorado (Page 3 line 23-25)'. After the 'Meanwhile', it begins to emphasize the role brown carbon played in the light absorption which has attracted increasing attention recently, as 'Meanwhile, the role of BrC in the radiation balance of the climate system has attracted increasing interest. Shamjad et al. (2015) mentioned that the atmospheric radiative forcing of total aerosol increases by 56%, and the forcing of BrC aerosol increases by 38% in the biomass burning season comparing to other seasons in Kanpur, India. Light absorption characteristics of WSOC aerosols are key factors in the climate forcing calculation (Page 3 line 26-29)'.

7. *Page 4, Line 2, why WSOC contribute to EC?*

Reply: The radiative forcing of WSOC doesn't contribute to that of EC. To eliminate the ambiguity of original expressions, the statement has been revised as '…that the WSOC radiative forcing accounted for 6.03 ± 3.62% and 11.41 ± 7.08% of EC radiative forcing at two stations there, respectively (Page 4 line 6-7)'.
The original expressions were '…that the relative contributions of radiative forcing of WSOC to EC were 6.03 ± 3.62% and 11.41 ± 7.08% at two stations there.'

8. *Page 4, Line 4, what are MAC365 and AAE*

Reply: $MAC_{365}$, also named $MAE_{365}$ is the mass absorption efficiency at 365 nm. MAE was explained as 'Mass absorption efficiency (MAE, $m^2$ $g^{-1}$) of BrC in WSOC was derived from Eq. (2):

$$MAE = b_{abs}/WSOC, \qquad (2)$$

where WSOC refers to the concentration of WSOC (μg $m^{-3}$) (Page 6 line 23-26)' and 'Mass absorption efficiency characterizes the efficiency of light absorption by WSOC. (Page 12 line 33)'.
AAE is Absorption Angstrom exponents which was explained as 'Absorption Angstrom exponents (AAE) were computed by Eq. (3):

$$b_{abs} \approx K \cdot \lambda^{-AAE}, \qquad (3)$$

where $K$ is a constant, $AAE$ describes the wavelength-dependent absorption enhancement of BrC, associated with its origin, size and composition (Bikkina, 2014); and $\lambda$ is wavelength in the range of 310-460 nm. AAE

for WSOC was fitted within the range of 310-460 nm in which coefficients of determination ($R^2$) for all samples are above 0.90 (Page 6 line 26-31)' and 'It is evident that the absorption coefficient of WSOC shows a relationship with wavelength, which can be described by absorption Angstrom exponent. AAE is also used to distinguish the BrC emission types due to its dependence on particle size and composition (Page 12 line 25-27)'.

In the original manuscript, $MAC_{365}$ and AAE in the Introduction were mentioned only as the examples of parameters related to the optical property, where the examples are not essential here. Furthermore, all of the relative optical parameters would be introduced in detail in latter statements mentioned above. Therefore, we removed these abbreviations in the introduction and revised the statements as 'The afore-mentioned method provides a rough estimation on the absorption of solar energy by WSOC and EC, by using the measured sample concentrations of WSOC and EC and the optical properties of WSOC. Many assumptions were applied in that method, thus uncertainties in the estimated forcing could be substantial (Page 4 line 7-9).'

The original statements were 'The afore-mentioned method provides a rough estimation on the absorption of solar energy by WSOC and EC, by using the measured sample concentrations of WSOC and EC and the optical properties of WSOC (MAC365 and AAE). Many assumptions were applied in that method (e.g., MAC and AAE of EC being constant), thus uncertainties in the estimated forcing could be substantial.'

9. *Page 10, the equations to estimate the contributions of biomass burning to carbonaceous components are based on the emissions of biomass burning. It should be discussed about the uncertainties as the ambient concentrations measured in Nanjing are mixture of biomass burning transported to here and local emissions from other sources.*

Reply: As suggested, the illustration about mentioned uncertainties were added at Page 11 line 3-5, as 'However, biomass burning contribution here could be related with both local emissions and long-range transport. The origins of biomass burning will be discussed in the following section.' More detailed discussion about the origin of biomass burning being a mixture of local emissions and long-range transport are in section 3.5.2 (Page 12 line 1-10), as 'In addition, there was a notable cluster originating from the coastal site of eastern China and mainly passing through Shanghai and Jiangsu, where no fire signals were found. But this cluster had a relatively high mean levoglucosan concentration (673 ng m$^{-3}$). According to the research by Zhou et al. (2017) on the biomass burning emission inventory, domestic straw burning was responsible for ~42 % of total biomass burning in China in 2012, and ~1.5 Gg and ~80 Gg $PM_{2.5}$ came

from domestic burning in Shanghai and Jiangsu, respectively. It illustrated the significant impact of indoor biomass burning which was invisible in MODIS fire spots map. Therefore, the levoglucosan enrichment of the cluster from coastal area might be due to domestic biomass burning. The air mass frequency of this cluster was 32.5%, the largest one among that for all clusters, which means domestic biomass burning in the eastern China much influenced aerosols in Nanjing. Local domestic biomass burning should be taken as an important factor to study aerosols of eastern China in further study, even though open biomass burning from long-range transport played a significant role in this study.'

10. *Page 11, the levoglucosan contration could be as high as 673 ngm-3 from non-biomass burning regions, you cannot put all 796 ngm-3 from southeast China to biomass burning.*

Reply: According to the revised statements in Page 12 line 7-10 'The air mass frequency of this cluster was 32.5%, the largest one among that for all clusters, which means domestic biomass burning in the eastern China much influenced aerosols in Nanjing. Local domestic biomass burning should be taken as an important factor to study aerosols of eastern China in further study, even though open biomass burning from long-range transport played a significant role in this study', indoor biomass burning which is invisible on MODIS fire spot map might result in the mean levoglucosan concentration of this cluster as high as 673 ng m$^{-3}$, which is supported by previous literature. Accordingly, we cannot deny that it is biomass burning attributing to this cluster originated from east China. Correspondingly, it is reasonable to attribute all 796 ng m$^{-3}$ from southeast China to biomass burning.

11. *Figure 10: from the relationship of OC, EC, and WSOC with lev, man, gal and nss-K in Figure 4, WSOC is correlated with biomass burning. Then, it is not clear why babs is not correlated with BB-WSOC at all (Figure 10), and it can not be explained by biomass burning is not the only source of BrC.*

Reply:   We revised Figure 10, presenting the correlation between babs and WSOC ($R^2 = 0.76$, $P < 0.01$) and the correlation between babs and WSOC from biomass burning, calculated with WSOC multiplied by BB-WSOC fraction ($R^2 = 0.55$, $P < 0.01$). Both correlations are significantly positive at 1% possibility level. But babs has a relatively higher $R^2$ with WSOC than WSOC from BB, indicating that other sources also contribute to water-soluble BrC, such as secondary aerosols.

The relative statements were revised as 'According to the analysis in Sect. 3.4, the robust correlations between WSOC and biomass burning tracers (levoglucosan, mannosan, galactosan or nss-K$^+$) implied that biomass burning has contributed to WSOC. With respect to this point, the correlation between $b_{abs}$ and WSOC ($R^2 = 0.76$, P < 0.01) and that between $b_{abs}$ and WSOC from biomass burning, calculated with WSOC multiplied by BB-WSOC fraction ($R^2 = 0.55$, P < 0.01), are presented in Fig. 10. Both correlations are significantly positive at 1% possibility level. But $b_{abs}$ has a relatively higher $R^2$ with WSOC than WSOC from BB, indicating that other sources also contributed to water-soluble BrC in the episode. Other researchers pointed out that secondary photochemical reactions in the atmosphere can produce water-soluble BrC as well (Sareen et al., 2013;Liu et al., 2016). Other researchers pointed out that secondary photochemical reactions in the atmosphere can produce water-soluble BrC as well (Sareen et al., 2013;Liu et al., 2016) (Page 12 line 18-24)'

Attached is revised figure 10 (Page 32):

[Figure]

**Figure 10: Correlations of water-soluble BrC absorption coefficient ($b_{abs}$) and WSOC or WSOC from biomass burning (WSOC from BB) in PM$_{2.5}$ from 14 to 29 Jan, 2015 in Nanjing. WSOC from BB is calculated with WSOC multiplied by BB-WSOC.**

12. *Contributions of biomass burning to carbonadoes aerosols are a key factor for analyzing their radiative effects. The authors should add these key information in abstract or conclusion.*

Reply: The description about biomass burning contributions to carbonaceous aerosols were added in abstract, as 'The biomass burning contribution to WSOC and OC was in the range of 1.1-55.4 % and 0.2-

 (Page 1 line 24-25)'. We also gave the relative information in the conclusion section (Page 14 line 15-16), stated as 'Biomass burning contribution to OC and WSOC was, on average, 20.9 ± 9.3 % and 22.3 ± 9.9 %, respectively, and was as high as 53.6 % to OC and 55.4 % to WSOC'.

13. *Figure 8 is not in high quality, where is Nanjing? I cann't agree with the authors that it shows "the air masses were mainly coming from the south and southeast regions".*

Reply: We are grateful to the reviewer for the comment. Taking the comment, we have remade Figure 8 in high resolution where Nanjing is clearly shown. The Flexpart simulation results indicated that the a notable fraction of air masses were coming from a region in the south and southeast of Nanjing. Accordingly, we have revised the sentence as "...the Flexpart potential emission sensitivity in Fig. 8 indicates that a notable fraction of air masses was coming from a region in the south and southeast of Nanjing (Page 11 line 29-30)"

14. *Table 2: I can understand that the authors want to compare with other studies, but I don't believe there are only four previous studies have reported lev concentration in the whole literature.*

Reply: More relevant researches were involved in Table 2 to give a more comprehensive description of levoglucosan levels in $PM_{2.5}$ under diverse types of environment.
Attached is the revised Table 2 (Page 37).

**Table 2. Comparison of levoglucosan concentration range in PM$_{2.5}$ during this study with those reported in the literature.**

| Sampling Site | Site Type | Sampling Time | Concentration Range (ng m$^{-3}$) | Reference |
|---|---|---|---|---|
| Azores, Portugal | coastal | Winter | 6.6 | |
| Puy de Dôme, France | coastal | Winter | 18.3 | (Puxbaum et al., 2007) |
| Schauinsland, Germany | coastal | Winter | 33.7 | |
| Sonnblick, Austria | coastal | Winter | 12.4 | |
| Summit, Greenland | coastal | May-December | 0.3 | (von Schneidemesser et al., 2009) |
| Jilin, China | forest | July | 42 (32-67) | |
| Shanghai, China | forest | June | 143 (20-212) | (Wang et al., 2008) |
| Guangdong, China | forest | August | 25 (0.3-61) | |
| Hainan, China | forest | November | 107 (19-398) | |
| Izana, Sahara | forest | August | 0.5 | (García et al., 2017) |
| Ovens, Australia | rural | Autumn | 870 | (Reisen et al., 2011) |
| Manjimup, Australia | rural | Autumn | 1060 | |
| Langley, Canada | rural | August | 26 | (Leithead et al., 2006) |
| Hok Tusi, Hong Kong | rural | Spring | 30 | (Sang et al., 2011) |
| Kowloon, Hong Kong | urban | Spring | 36 | |
| National University, Singapore | urban | September | 91.2 | (Yang et al., 2013) |
| Grenoble, France | urban | January | 815 | (Favez et al., 2010) |
| Beijing, China | urban | Summer | 230 | |
| | | BB episode | 750 | |
| | | Typical summer | 120 | |
| | | Winter | 590 | (Cheng et al., 2013) |
| | | Firework episode | 460 | |
| | | Typical winter | 640 | |
| Shanghai, China | urban | Spring | 66 (18-159) | |
| | | Summer | 28 (8.6-194) | |
| | | Autumn | 229 (13-1606) | (Li et al., 2016) |
| | | Winter | 161 (26-614) | |
| Nanjing, China | urban | Winter | 373 (22.4-1476) | Current study |

Anonymous Referee #2

*The study quantitatively evaluated the influence of long-range transport of biomass burning (BB) emissions on carbonaceous aerosols in Nanjing city, China, which is helpful to understand the sources and associated contributions to aerosol carbonaceous component in urban regions in China. Following issues need to be addressed prior to publication.*

1. *One concern is about the relative contributions of long-range transport BB versus the local domestic BB emissions shown in Fig. 7. Though the levoglucosan concentration contained in air masses from southeast China is higher than that from the coastal region of eastern China, the air mass frequency from the latter air mass cluster (32.5%) is much higher than the former one (20%). After weighted with the cluster frequency, I would say the contribution from coastal eastern China actually is larger than that from southeast China. I understand that the long-range BB could contribute to aerosols in Nanjing significantly during certain time period, but, from long-term perspective, it might not be the case due to the large contribution from local domestic BB emissions. As such, the authors may want to make changes on the statements related to the contribution of long-range contribution.*

Reply: As suggested, we have revised the statements (Page 12 line 1-2, line 7-10). The revised statements along with original statements in Page 12 line 2-7 make a more comprehensive discussion about the contribution of long-range transport and the potential impacts from domestic biomass burning emissions in this study. The statements after revision are as below 'In addition, there was a notable cluster originating from the coastal site of eastern China and mainly passing through Shanghai and Jiangsu, where no fire signals were found. But this cluster had a relatively high mean levoglucosan concentration (673 ng m$^{-3}$). According to the research by Zhou et al. (Zhou et al., 2017) on the biomass burning emission inventory, domestic straw burning was responsible for ~42 % of total biomass burning in China in 2012, and ~1.5 Gg and ~80 Gg PM$_{2.5}$ came from domestic burning in Shanghai and Jiangsu, respectively. It illustrated the significant impact of indoor biomass burning which was invisible in MODIS fire spots map. Therefore, the levoglucosan enrichment of the cluster from coastal area might be due to domestic biomass burning. The air mass frequency of this cluster was 32.5%, the largest one among that for all clusters, which means domestic biomass burning in the eastern China much influenced aerosols in Nanjing. Local domestic biomass burning should be taken as an important factor to study aerosols of eastern China in further study, even though open biomass burning from long-range transport played a significant role in this study.'

The original relative statements were 'Notably, there is a cluster originating from the coastal site of eastern China mainly passing through Shanghai and Jiangsu, where no fire signals are found. But this cluster has a relatively high mean levoglucosan concentration (673 ng m$^{-3}$). According to the research by Zhou et al. (Zhou et al., 2017) on biomass burning emission inventory, domestic straw burning was responsible for ~42 % of total biomass burning in China in 2012, and ~1.5 Gg and ~80 Gg PM$_{2.5}$ came from domestic

2. *The physical and optical properties of black carbon after emitted into atmosphere for certain time can be largely modified by the aging process (e.g., Peng et al., PNAS, 2016; Wang et al., JAMES, 2018). Please add some reviews regarding soot aging in the introduction section on page 4 when reviewing the alterations of physical and optical properties with the transport process.*

Reply: As suggested, we added review about soot aging in the introduction as 'Using a novel chamber method, Peng et al., (2016) found two stages in the aging of BC: initial transformation with little absorption variation and subsequent growth with a profound absorption enhancement. The fact was also revealed that BC under more polluted urban background had a more remarkable impact on the pollution development and the radiative forcing enhancement (Peng et al., 2016). Model simulations calibrated by chamber measurement showed significant differences in BC burden and radiative forcing from the simulations without considering BC aging. In addition, BC coating materials from aging processes were found responsible for a net increase of BC radiative forcing (Wang et al., 2018b) (Page 4 lines 21-27)'.

3. *In abstract: when talking about absorption of carbonaceous aerosols, the two terms (i.e., WSOC and water soluble BrC) are basically equivalent, but the authors switched back and forth to use them. Please stick to one expression.*

Reply: As suggested, water-soluble BrC has replaced WSOC when talking about the radiative absorption in abstract (Page 2 line 1-3). The revised statements are 'The solar energy absorption of water-soluble BrC due to biomass burning was estimated as $0.2 \pm 0.1$ (0.0-0.9) W m$^{-2}$. Potential Source Contribution Function model simulations showed that the solar energy absorption induced by water-soluble BrC and EC aerosols was mostly due to the regional transported carbonaceous aerosols from the source regions such as southeast China.'

The original statements were 'The solar energy absorption of WSOC due to biomass burning was estimated as $0.2 \pm 0.1$ (0.0-0.9) W m$^{-2}$. Potential Source Contribution Function model simulations showed that the solar energy absorption induced by WSOC and EC aerosols was mostly due to the regional transported carbonaceous aerosols from the source regions such as southeast China.'

4. *Since the observations are ground-based, while the back-trajectory analysis focuses on 500m level. Are the conclusions about relative contributions from different air mass clusters sensitive to the selected altitude?*

Reply: Considering that the planetary boundary layer usually has a lower height in wintertime, 500 m is a proper height set as the planetary boundary layer height, which is universal in other study (Yang et al., 2017).

5. *Typos Line 26 page 2: missing a preposition at ". . .frequently method biomass burning. . ."*

Reply: The statement has been revised as 'In the observation-based studies, the characteristic ratio calculation and positive matrix factorization (PMF) are frequently methods to calculate biomass burning contribution (Page 2 line 29-31)'.

The original statement was 'In the observation-based studies, the characteristic ratio calculation and positive matrix factorization (PMF) are the most frequently methods biomass burning contribution.'

Anonymous Referee #3

*The manuscript measured the chemical and optical properties of carbonaceous aerosols in Nanjing and attempted to elaborate impacts of the biomass burning (BB) originated from regional transport on the carbonaceous aerosols. The topic is of interest but the manuscript is not well written. There are several factors hinder publication of the manuscript at the present form, and a major revision is needed.*

Major comments:

1. *The title is "Impact of regional-transported biomass burning emissions on chemical and optical properties of carbonaceous aerosols in Nanjing, East China". Therefore, It is assumed that the measured aerosol properties in Nanjing with and without BB impact should have large difference. Figure 5 shows that there are moments with low BB impact on Nanjing. Did the chemical and optical properties of carbonaceous aerosols under the low BB influence reveal a significant difference from those under the high BB influence?*

Reply: As suggested, T-test method has been applied to test the significance of chemical and optical properties difference between samples with high BB influence (BB-OC > 20.9%) and samples with low BB influence (BB-OC < 20.9%), where 20.9% is the mean BB-OC value of the event in this study. The T-test results shows that all of carbonaceous components (OC, EC, WSOC), BB tracers (lev, man, gal) and $b_{abs}$ of water-soluble BrC exhibits a significant difference between samples under high and low BB influence (h = 1, p < 0.05). For a better representative of the study, we changed the title as "==Chemical and Optical Properties of Carbonaceous Aerosols in Nanjing, East China: Regional-transported Biomass Burning Contribution== (Page 1 line 1-3)".
Attached table shows the T-test results:

|     | OC     | EC     | WSOC   | levoglucosan | mannanson | galactosan | $b_{abs}$ |
| --- | ------ | ------ | ------ | ------------ | --------- | ---------- | --------- |
| h   | 1      | 1      | 1      | 1            | 1         | 1          | 1         |
| sig | 0.0111 | 0.0083 | 0.0022 | 0.0000       | 0.0000    | 0.0000     | 0.0004    |

2. *Figure 5 shows that the fraction of BB-OC and BB-WSOC is nearly the same throughout the measurement. Does this mean that the emission ratios of WSOC/OC from other anthropogenic sources are the same as BB, or is it the coincidence from the calculation by the equation 6 and 7?*

Reply: We agree that the same trends and the good correlation of BB-OC and BB-WSOC in Figure 5 were due to the calculation coincidence.
However, even though the reference parameters for BB-OC and BB-WSOC calculation are from two

$$BB - OC = \frac{(lev/(1000 \times OC))_{ambient}}{0.082} \times 100\%, lev: ng\ m^{-3}; OC: \mu g\ m^{-3} \qquad (6)$$

(Zhang et al., 2010;Puxbaum et al., 2007;Zdráhal et al., 2002; Sang et al., 2011). In biomass burning source emission tests for three major types of cereal straw (corn, wheat, and rice) in China, 0.082 was reported as the lev/OC ratio for PM$_{2.5}$ (Zhang et al., 2007). This value can be used in combination with the lev/OC ratios of our PM$_{2.5}$ samples to roughly estimate the contribution of biomass burning smoke to the ambient OC.'

The statements about BB-WSOC calculations are in Page 10 line 27-31, as 'The contribution from biomass burning to the WSOC in PM2.5 (BB-WSOC) trend in Fig. 5 makes the above analysis more convincing due to the consistency of the contributions computed by two different methods. BB-WSOC was estimated with Eq. (7):

$$BB - WSOC = \frac{(lev/WSOC)_{ambient}}{0.17} \times 100\%, lev/WSOC: \mu g\ \mu gC^{-1} \qquad (7)$$

which used a lev/WSOC ratio of 0.17 µg µgC-1 from the test burns of rice straws and wheat straw (Yan et al., 2015)'

Attached is Figure 5 (Page 27)

[Figure]

**Figure 5: Time series of biomass burning contribution to OC (BB-OC) and WSOC(BB-WSOC).**

3. *Since the authors attempted to investigate the impact of BB on carbonaceous aerosols, why did the authors only measure the absorption of BrC in WSOC, instead of the absorption of total BrC in OC? There are also water-insoluble BrC from BB.*

Reply: First, WSOC accounted for averaged 50% and up to 90% of OC and showed a stronger correlation with light-absorbing property (b$_{abs}$) than OC in our study (R$^2$ = 0.76 for WSOC and R$^2$ = 0.72 for OC, p < 0.01). Additionally, previous studies showed that biomass burning was one of the predominant sources of WSOC and water-soluble BrC (Hoffer et al., 2006). Therefore, it's of great significance to discuss the light-

Hoffer, A., Gelencsér, A., Guyon, P., Kiss, G., Schmid, O., Frank, G. P., Artaxo, P., and Andreae, M. O.: Optical properties of humic-like substances (HULIS) in biomass-burning aerosols, Atmos. Chem. Phys., 6, 3563-3570, 10.5194/acp-6-3563-2006, 2006.

Liu, J., Bergin, M., Guo, H., King, L., Kotra, N., Edgerton, E., and Weber, R. J.: Size-resolved measurements of brown carbon in water and methanol extracts and estimates of their contribution to ambient fine-particle light absorption, Atmos. Chem. Phys., 13, 12389-12404, 10.5194/acp-13-12389-2013, 2013.

4. *Page 8, Line 17, the authors concluded that secondary OC is weak and stable only based on the OC/EC ratio ranging from 2 to 4 during the severe pollution period. The conclusion is arbitrary. Firstly, the OC/EC ratio ranging from 2 to 4 are rather large. Secondly, previous studies have indicated that SOA contribution during haze days during wintertime in China is considerable. For example, Huang et al. (2014) showed SOA contributed 44%-71% to the OA mass in four megacities of China during wintertime. Sun et al. (2013) revealed that SOA contributed 31% to the OA mass in Beijing in winter.*

Reply: As suggested, we revised the statements and added the references to describe the results more objectively, as 'As the pollution got more severe over time, it shrank to 2.0-4.0 after 16 Jan, indicating that the secondary sources had a relatively stable but still considerable impact on $PM_{2.5}$ at the polluted stage. According to the previous studies, secondary organic aerosol (SOA) usually played an important role in the wintertime air pollution (Huang et al., 2014;Sun et al., 2013) (Page 8 line 22-25)'.

The original statements were 'In contrast to combustion sources, the secondary sources had a relatively weak and stable impact on $PM_{2.5}$ at the polluted stage.'

Minor comments:

5. *Page 2, Lines 11-15: What is the difference among those portions of OC?*

Reply: Each portion of OC has different size distributions, chemical molecular compositions or physical morphology (i.e. fractal or spherical) leading to the different optical properties. Here, we introduced a relative reference in the text (Page 2 line 15). According to the literature, this classification depends on the optical property of OC, as stated in text 'Meanwhile, a portion of organic carbon (OC) could yield a cooling effect on the land surface by scattering sunlight (Zhang et al., 2017; Myhre et al., 2013). A portion of OC involves in the aging process of elemental carbon (EC) or black carbon (BC) as coating materials for the BC core, enhancing BC radiative absorption (Peng et al., 2016; Wang et al., 2018b). A fraction of OC can also directly absorb solar energy functioning similarly as BC, which is referred to as brown carbon (BrC) (Laskin et al., 2015). But BrC is different from BC due to its strong light absorption from visible (VIS) to ultraviolet (UV) wavelengths, thus having a stronger wavelength dependence than BC (Andreae and Gelencsér, 2006) (Page 2 line 11-17)'.

17), only describing the light absorbing characteristics of BrC which is different from BC. The estimated radiative forcing of BC and BrC in other literature were given following this sentence that 'Radiative forcing of BC is estimated to be 0.2-1.2 W m-2 (Moffet and Prather, 2009), while radiative forcing of BrC contribution is suggested to be up to 0.25 W m-2 or to contribute to 19 % of the total atmospheric absorption computed by model simulations (Feng et al., 2013) (Page 2 Line 17-19)', showing that BC makes a stronger radiative forcing than BrC, but BrC absorption is non-negligible.

7. *Page 2, Line 26:  what is "the most frequently method"?*

Reply: The statement has been revised to eliminate the expression ambiguity as 'In the observation-based studies, the characteristic ratio calculation and positive matrix factorization (PMF) are frequently used methods to calculate biomass burning contribution (Page 2 line 29-31)'. Both the characteristic ratio calculation and positive matrix factorization (PMF) are frequently methods to calculate the biomass burning contribution, which were applied in this study.
The original statements were 'In the observation-based studies, the characteristic ratio calculation and positive matrix factorization (PMF) are the most frequently methods biomass burning contribution.'

8. *Page 3, Line 3, "that" should be removed?*

Reply: It has been revised to as suggested (Page 3 line 7-8).

9. *Page 3, Line 18, "impact" should read "impacts".*

Reply: It has been revised as suggested (Page 3 line 22).

10. *Page 3, Line 21: "than" should read "compared to".*

Reply: It has been revised as suggested (Page 3 line 25).

11. *Page 3, Lines 25-27:  Are the three parameters are specially used for BrC from BB?*

Reply: These three parameters are main parameters to describe optical properties of both BrC and EC from all kinds of sources. Research about biomass burning involving these parameters are listed afterwards, for our study focuses on the impact of biomass burning. We removed the misleading statements. The revised statements are 'Light-absorbing parameters including absorption coefficient, mass absorption efficiency and absorption Angstrom exponent, have been involved to describe the light absorption characteristics of BrC (Alexander et al., 2008;Yang et al., 2009;Chakrabarty et al.,

calculated mass absorbance efficiency for WSOC from a biomass burning factor derived from positive matrix factorization (PMF) receptor model. It showed that the biomass burning source contributed to 58% of total light absorption at 365 nm and WSOC associated with sulfate and oxalate contributed to 21% (Page 3 line 29-Page 4 line 3).'

*12. Page 3, Lines 28: What is HULIS?*

Reply: The full name of HULIS, 'humic-like substances' has been added where HULIS was first presented (Page 3 line 32).

*13. Page 7, Line5, "give" should be "given".*

Reply: It has been revised as suggested (Page 7 line 12).

*14. Page 8, Line 10, "uncertainty" should be "variation"*

Reply: It has been revised as suggested (Page 8 line 16).

*15. Page 10, Line 25, "which illustrating the necessity of parameter BB-OC". The sentence is hard to understand.*

Reply: We removed the sentences after Page 10 line 26, which were not closely related to the main idea of this paragraph.

The removed sentences are ' There were three remarkable BB-OC peaks at 08:00 on 20 Jan, 14:00 on 21 Jan and 08:00 on 25 Jan in Fig. 5, indicating the large contributions of biomass burning during those periods. However, EC and biomass burning tracers were not rich at 08:00 on 20 Jan, implying that combustion activities were not highly intense at that time. But it doesn't contradict the fact that biomass burning made a significant contribution to OC compared to other pollution sources at 08:00 on 20 Jan. It demonstrates that it's very necessary to use BB-OC to depict the contribution of biomass burning instead of biomass burning tracer levels, for BB-OC excludes the influence of the OC level. '

[revised manuscript text omitted]

---

## Author Comment (AC2) · 17 May 2019

The comment was uploaded in the form of a supplement:
https://www.atmos-chem-phys-discuss.net/acp-2018-1029/acp-2018-1029-AC2-supplement.pdf

---

## Author Response (AR1)

**RE: A point-to-point response to reviewers' comments**

'Impacts of Regional-transported Biomass Burning Emissions on Chemical and Optical Properties of Carbonaceous Aerosols in Nanjing, East China' (acp-2018-1029) by Xiaoyan Liu, Yan-Lin Zhang, Yiran Peng, Lulu Xu, Chunmao Zhu, Xiaoyao Zhai, Chi Yang, Mozammel Haque, Fang Cao, Yunhua Chang, Tong Huang, Zufei Xu, Mengying Bao, Wenqi Zhang, Meiyi Fan, Xuhui Lee

We are grateful to the helpful comments by the referees, and have carefully revised our manuscript accordingly. A point-to-point response to the reviewers' comments is given below, in which comments are repeated in italic; the citations from the revised manuscript are underlined; the revised or added statements are highlighted in yellow. The marked-up manuscript version is attached after the point-to-point response, in which the changes made are highlighted in yellow.

**Anonymous Referee #1**

*This study investigates the chemical and optical properties of carbonaceous aerosols in Nanjing and finds that the transport of biomass burning emissions is a major source of carbonaceous aerosols. The paper is clear on describing the results they found, and the conclusion is solid. My largest concern is the way how the authors present the paper. From my understanding, the biomass burning emissions are transported to atmosphere in Nanjing and they showed different properties than other episodes, it is not the case that the transported emission changed the properties of aerosols emitted in Nanjing. Thus, I would avoid saying that the "Impacts of Regional-transported Biomass Burning Emissions on", which is misleading.*

Reply: We are grateful for the kindly comments from the referee. In regards to the suggestion about the title, here's our reply:

In order to avoid being misleading, we would like to revise the title as 'Chemical and Optical Properties of Carbonaceous Aerosols in Nanjing, East China: Regional-transported Biomass Burning Contribution (Page 1 line 1-3)'.

We agree with the reviewer in the sense that aerosols collected in situ observations are affected by both local emissions and regional transported emissions. This study dedicated to figure out the role of regional transported biomass burning emissions in the pollution episode and it turned out that the chemical and optical properties of ambient aerosols in Nanjing was influenced by regional-transported biomass burning emissions during the episode because of the mixing of aerosols with diverse origins. In such a sense, we put 'aerosols in Nanjing' which directs general ambient aerosols in Nanjing atmosphere, instead of 'aerosols from local emissions of Nanjing'.

Detailed points:

1. *Page 1: Abstract: "Biomass burning can significantly impact the chemical and optical properties of carbonaceous aerosol", this sentence is repeating with the next one.*

Reply: As suggested, we removed the duplicated expressions and revised the description in Abstract as '==Biomass burning can significantly impact the chemical and optical properties of carbonaceous aerosols. Here, the biomass burning impacts were studied during wintertime in a megacity of Nanjing, East China== (Page 1 line 20-21)'.

The original sentences were 'Biomass burning can significantly impact the chemical and optical properties of carbonaceous aerosols. Here, the impacts of biomass burning emissions on chemical and optical properties of carbonaceous aerosols were studied during wintertime in a megacity of Nanjing, East China.'

*2. Page 2, Lines 5-6, please avoid repeating such blurry information.*

Reply: As suggested, we deleted the original sentence 'The regional transported biomass burning emissions may significantly impact the chemical and optical properties of carbonaceous aerosols in the polluted atmosphere', which indefinitely attributed the impacts of aerosols chemical and optical property to transported biomass burning emissions.

*3. Page 2, Lines 21: long-range transport of aerosols; "simulation modeling"?*

Reply: The impacts due to biomass burning are regional and global owing to the aerosols long-range transport, as stated in the original manuscript that 'All effects mentioned above are both regional and global owing to the aerosols long-range transport (Page 2 line 21-22)'. In order to figure out the role of long-range transport in the aerosol pollution events, numerical simulations were applied in previous studies, such as the references mentioned in our revised manuscript that '==Studies based on numerical simulations have shown that emissions from biomass burning can be transported to remote sites, even across the oceans (Aouizerats et al., 2015; Zhu et al., 2016; Ancellet et al., 2016)== (Page 2 line 22-23)'. Additionally, a clear statement including more details about numerical simulations research cases were added in our revised manuscript as '==Aouizerats et al. (2015) modeled a large fire-induced haze episode in 2006 stemming mostly from Indonesia using the Weather Research and Forecasting model coupled with Chemistry and found out the notable impact of long-range transported aerosols from Indonesia on ambient air quality and climate in Singapore. Ancellet et al. (2016) made the assessment of aerosol transport from North America to Europe using forward simulations of the FLEXPART Lagrangian particle dispersion model== (Page 2 line 23-27).'

*4. Page 2, Lines 26: methods for biomass burning calculation*

Reply: The expression mistake has been corrected. The revised statements are '==In the observation-based studies, the characteristic ratio and positive matrix factorization are frequently used to calculate biomass burning contribution== (Page 2 line 29-30)'.

The original statements were 'In the observation-based studies, the characteristic ratio calculation and positive matrix factorization (PMF) are the most frequently methods biomass burning contribution.'

*5. Page 3, Line 20, POM?*

Reply: 'POM' has been replaced by its full name '==primary organic matter==' (Page 3 line 24-25).

*6.  Page 3, Line 21, This is no turning here to use "However"*

Reply: As suggested, 'However' has been replaced by 'Meanwhile' which better indicates the similar climatic effects between BC and BrC and helps to switch the emphasis from black carbon to brown carbon to further discuss the significant impact of brown carbon on radiative process. This paragraph is discussing the biomass burning aerosols impact on radiative forcing. Before the 'Meanwhile', it is describing the radiative forcing enhancement by black carbon, as 'Biomass burning aerosol has significant impacts on regional and global radiative forcing (Ramanathan and Carmichael, 2008). There is no doubt that elemental carbon increases solar absorption of ambient aerosols most effectively (Ramanathan and Carmichael, 2008; Myhre and Samset, 2015; Li et al., 2016b). Lack et al. (2012) found that the absorptivity of BC and primary organic matter mixture has 70 % enhancement in biomass burning period compared to normal days in Boulder, Colorado (Page 3 line 22-25)'. After the 'Meanwhile', it begins to emphasize the role brown carbon played in the light absorption which has attracted increasing attention recently, as 'Meanwhile, the role of BrC in the radiation balance of the climate system has attracted increasing interest. Shamjad et al. (2015) mentioned that the atmospheric radiative forcing of total aerosol increases by 56 %, and the forcing of BrC aerosol increases by 38 % in the biomass burning season comparing to other seasons in Kanpur, India. Light absorption characteristics of WSOC aerosols are key factors in the climate forcing calculation (Page 3 line 26-29)'.

*7.  Page 4, Line 2, why WSOC contribute to EC?*

Reply: The radiative forcing of WSOC doesn't contribute to that of EC. We'd like to compare values of radiative forcing caused by WSOC and EC. To eliminate the ambiguity of original expressions, the statement has been revised as '…that the radiative forcing caused by WSOC was 6.03 ± 3.62 % and 11.41 ± 7.08 % of that caused by EC at two stations there, respectively (Page 4 line 5-6)'.
The original expressions were '…that the relative contributions of radiative forcing of WSOC to EC were 6.03 ± 3.62% and 11.41 ± 7.08% at two stations there.'

*8.  Page 4, Line 4, what are MAC365 and AAE*

Reply: $MAC_{365}$, also named $MAE_{365}$ is the mass absorption efficiency at 365 nm. MAE was explained as 'Mass absorption efficiency (MAE, $m^2\ g^{-1}$) of BrC in WSOC was derived from Eq. (2):

$$MAE = b_{abs}/WSOC, \qquad (2)$$

where WSOC refers to the concentration of WSOC ($\mu g\ m^{-3}$) (Page 6 line 23-26)' and 'Mass absorption efficiency characterizes the efficiency of light absorption by WSOC (Page 13 line 3)'.
AAE is Absorption Angstrom exponents which was explained as 'Absorption Angstrom exponents (AAE) were computed by Eq. (3):

$$b_{abs} \approx K \cdot \lambda^{-AAE}, \qquad (3)$$

where $K$ is a constant, $AAE$ describes the wavelength-dependent absorption enhancement of BrC, associated with its origin, size and composition (Bikkina, 2014); and $\lambda$ is wavelength in the range of 310-460 nm. AAE for WSOC was

fitted within the range of 310-460 nm in which coefficients of determination ($R^2$) for all samples are above 0.90 (Page 6 line 26-31)' and 'It is evident that the absorption coefficient of WSOC shows a relationship with wavelength, which can be described by absorption Angstrom exponent. AAE is also used to distinguish the BrC emission types due to its dependence on particle size and composition (Page 12 line 29-31)'.

5    In the original manuscript, $MAC_{365}$ and AAE in the Introduction were mentioned only as the examples of parameters related to the optical property, where the examples are not essential here. Furthermore, all of the relative optical parameters would be introduced in detail in latter statements mentioned above. Therefore, we removed these abbreviations in the introduction and revised the statements as 'The afore-mentioned method provides a rough estimation on the absorption of solar energy by WSOC and EC, by using the measured sample concentrations of

10    WSOC and EC and the optical properties of WSOC. Many assumptions were applied in that method, thus uncertainties in the estimated forcing could be substantial (Page 4 line 6-9).'

The original statements were 'The afore-mentioned method provides a rough estimation on the absorption of solar energy by WSOC and EC, by using the measured sample concentrations of WSOC and EC and the optical properties of WSOC (MAC365 and AAE). Many assumptions were applied in that method (e.g., MAC and AAE of EC being

15    constant), thus uncertainties in the estimated forcing could be substantial.'

9.   *Page 10, the equations to estimate the contributions of biomass burning to carbonaceous components are based on the emissions of biomass burning. It should be discussed about the uncertainties as the ambient concentrations measured in Nanjing are mixture of biomass burning transported to here and local emissions from other sources.*

Reply: Here, in equation (6) and (7) in Page 10-11, *OC* and *WSOC* refer to their ambient concentrations from all kinds of sources; *Lev* refers to ambient levoglucosan concentration from biomass burning. Notably, all the sources including biomass burning could be involved with both local emissions and long-range transport to Nanjing, as the ambient aerosols are usually with a mix of origins due to the atmospheric motion. Therefore, the BB-OC and BB-

25    WSOC are objectively describing the contribution from biomass burning regardless of its origins.

But we agree to clarify that the calculated results from equation (6) and (7) indicate the total contributions of biomass burning in both local and remote area. As suggested, the corresponding illustrations were added at Page 11 line 6-8, as 'However, biomass burning contribution here could be related with both local emissions and long-range transport due to the atmospheric motion and the subsequent mixing of aerosols. The origins of biomass burning will

30    be discussed in the following section.'

10.   *Page 11, the levoglucosan contration could be as high as 673 ngm-3 from non-biomass burning regions, you cannot put all 796 ngm-3 from southeast China to biomass burning.*

35    Reply: Firstly, compared to non-biomass burning regions, we prefer to classify the regions that cluster (673 ng m$^{-3}$) passing by in Fig.7 as the non-open biomass burning regions. No visible fire spots can only exclude open biomass burning in these regions. There might still be domestic biomass burning in these regions which are invisible on the MODIS fire spots map. Considering that the methods of assessing domestic biomass burning emissions are limited now, we are not able to accurately distinguish the domestic and open biomass burning emissions in this study, but

40    only able to give the related previous results based on inventory investigations from other studies as a supportive evidence, that 'According to the research by Zhou et al. (2017) on the biomass burning emission inventory, domestic

straw burning was responsible for ~42 % of total biomass burning in China in 2012, and ~1.5 Gg and ~80 Gg $PM_{2.5}$ came from domestic burning in Shanghai and Jiangsu, respectively. It illustrated the significant impact of indoor biomass burning which was invisible in MODIS fire spots map because satellite can only detect open biomass burning. Therefore, the levoglucosan enrichment of the cluster from coastal area might be due to domestic biomass burning (Page 12 line 7-11).' Moreover, we revised the relative statements to underline the role domestic biomass burning played in the episode, as 'The air mass frequency of this cluster was 32.5 %, the largest one among that for all clusters, which means domestic biomass burning in the eastern China much influenced aerosols in Nanjing. Domestic biomass burning should be taken as an important factor to study aerosols of eastern China in further study, even though open biomass burning from long-range transport played a significant role in this study (Page 12 line 11-14).'

In addition, the values of levoglucosan concentration shown in Fig.7 (such as 673 ng $m^{-3}$ and 796 ng $m^{-3}$) were calculated 'in order to compare the relative levels of biomass burning emissions transported by the four clusters (added in Page 11 line 30)', no matter these biomass burning emissions were open or domestic. Therefore, it's reasonable to attribute the whole levoglucosan concentration of 796 ng $m^{-3}$ to the biomass burning and also to the long- range transport originating from southeastern China. To make it better understood, we added and revised the relative statements, as 'All 48 h backward air mass trajectories were classified into four clusters and mean levoglucosan concentrations of all trajectories for each cluster are described in Fig. 7 in order to compare the relative levels of biomass burning emissions transported by the four clusters. The cluster originating from southeastern China had the highest mean levoglucosan concentration (796 ng $m^{-3}$) among the four clusters, although it transported the least air mass to Nanjing (20 %). Moreover, when the maximum levoglucosan was observed between 05:00 and 08:00 on 25 Jan, the FLEXPART potential emission sensitivity in Fig. 8 indicates that a notable fraction of air masses was coming from a region in the south and southeast of Nanjing. It is evident that air masses caught the pollutants emitted from biomass burning on 21-24 Jan, as was illustrated by the hotspots. Therefore, the biomass burning emissions from southeastern China profoundly impacted the aerosols in Nanjing during the episode (Page 11 line 28-Page 12 line 4).'

Attached is Figure 7 in Page 32 for reference,

[Figure]

**Figure 7: Air mass backward trajectory from 21 to 25 Jan, along with the MODIS fire spot map. Percentage refers to the air mass contribution to Nanjing and values with a unit of ng $m^{-3}$ refers to mean concentration of levoglucosan for every trajectory in that cluster.**

11. *Figure 10: from the relationship of OC, EC, and WSOC with lev, man, gal and nss-K in Figure 4, WSOC is correlated with biomass burning. Then, it is not clear why babs is not correlated with BB-WSOC at all (Figure 10), and it can not be explained by biomass burning is not the only source of BrC.*

5   Reply:  We revised Figure 10, presenting the correlation between $b_{abs}$ and WSOC ($R^2 = 0.76$, P < 0.01) and the correlation between $b_{abs}$ and WSOC from biomass burning, calculated with WSOC multiplied by BB-WSOC fraction ($R^2 = 0.55$, P < 0.01). Both correlations are significantly positive at 1% possibility level. But $b_{abs}$ has a relatively higher $R^2$ with WSOC than WSOC from BB, indicating that other sources also contributed to water-soluble BrC, such as secondary organic aerosols.

10  The relative statements were revised as 'According to the analysis in Sect. 3.4, the robust correlations between WSOC and biomass burning tracers (levoglucosan, mannosan, galactosan or nss-K$^+$) imply that biomass burning had contributed to WSOC. With respect to this point, the correlation between $b_{abs}$ and WSOC ($R^2 = 0.76$, P < 0.01) and that between $b_{abs}$ and WSOC from biomass burning, calculated with WSOC multiplied by BB-WSOC fraction ($R^2 = 0.55$, P < 0.01), are presented in Fig. 10. Both correlations are significantly positive at 1 % possibility level.

15  But $b_{abs}$ has a relatively higher $R^2$ with WSOC than WSOC from BB, indicating that other sources also contributed to water-soluble BrC in the episode. Other researchers pointed out that secondary photochemical reactions in the atmosphere can produce water-soluble BrC as well (Sareen et al., 2013; Liu et al., 2016) (Page 12 line 22-28).'
The attached is revised Figure 10 (Page 35):

20  **Figure 10: Correlations of water-soluble BrC absorption coefficient ($b_{abs}$) and WSOC or WSOC from biomass burning (WSOC from BB) in PM$_{2.5}$ from 14 to 29 Jan, 2015 in Nanjing. WSOC from BB is calculated with WSOC multiplied by BB-WSOC.**

12. *Contributions of biomass burning to carbonadoes aerosols are a key factor for analyzing their radiative effects. The authors should add these key information in abstract or conclusion.*

Reply: The description about biomass burning contributions to carbonaceous aerosols were added in abstract, as 'The significant contribution of biomass burning to water-soluble organic carbon, WSOC ($22.3 \pm 9.9$ %) and organic carbon, OC ($22.3 \pm 9.9$ %) was observed in this study (Page 1 line 24-25)' and 'The solar energy absorption of water-soluble BrC due to biomass burning was estimated as $0.2 \pm 0.1$ (0.0-0.9) W m$^{-2}$, considering the biomass

30  burning contribution to carbonaceous aerosols (Page 2 line 1-2)' We also gave the relative information in the conclusion section, stated as 'Biomass burning contribution to OC and WSOC was, on average, $20.9 \pm 9.3$ % and $22.3 \pm 9.9$ %, respectively, and was as high as 53.6 % to OC and 55.4 % to WSOC, reflecting the large contribution of biomass burning to carbonaceous aerosols in urban atmosphere during winter (Page 14 line 20-22)'.

*13. Figure 8 is not in high quality, where is Nanjing? I cann't agree with the authors that it shows "the air masses were mainly coming from the south and southeast regions".*

5   Reply: We are grateful to the reviewer for the comment. Taking the comment, we have remade Figure 8 in high resolution where Nanjing is clearly shown (Page 33). The FLEXPART simulation results indicated that a notable fraction of air masses was coming from a region in the south and southeast of Nanjing. Accordingly, we have revised the sentence as "...the FLEXPART potential emission sensitivity in Fig. 8 indicates that a notable fraction of air masses was coming from a region in the south and southeast of Nanjing ( Page 12 line 1-2)"

10   Attached is the revised Figure 8 (Page 33).

[Figure]

Four-day PES of BC at NUPT in Jan 25 05:00 - 08:00 (LT), 2015

**Figure 8: The four-day footprint of NUPT (118.78° E, 32.09° N, black point), the observation site, starting from 05:00-08:00 LT on 25 Jan, 2015 when the maximum levoglucosan was observed. The footprint was illustrated as FLEXPART potential emission sensitivity shown as the retention time in each grid at 0–500 m contributing to the observed elemental carbon.**

*14. Table 2: I can understand that the authors want to compare with other studies, but I don't believe there are only four previous studies have reported lev concentration in the whole literature.*

Reply: More relevant researches were involved in Table 2 to give a more comprehensive description of levoglucosan

20   levels in $PM_{2.5}$ under diverse types of environment. Accordingly, related statements have been revised as 'The average concentration of levoglucosan in $PM_{2.5}$ was 373 ± 268 ng m$^{-3}$ in this study, which was significantly higher than those observed at the coastal sites in Europe (Puxbaum et al., 2007; von Schneidemesser et al., 2009), the forest sites in the eastern China (Wang et al., 2008), and some rural sites in Canada and Hong Kong (Leithead et al., 2006; Sang et al., 2011). But for the rural sites of Australia, levoglucosan was found much more enriched than in this study

25   (Reisen et al., 2011). For the urban areas, levoglucosan level in this study was higher than the level in Shanghai, Hong Kong, Singapore, as well as the southeastern US (Li et al., 2016; Sang et al., 2011; Yang et al., 2013; Zhang et al., 2010a), but was lower than those measured in Grenoble, France and also Beijing during winter and a biomass burning episode in summer (Cheng et al., 2013; Favez et al., 2010) (Table 2) (Page 9 line 25-32).'

Attached is the revised Table 2 (Page 40).

**Table 2. Comparison of levoglucosan concentration level in PM$_{2.5}$ during this study with those reported in the literature.**

| Sampling Site | Site Type | Sampling Time | Concentration (ng m$^{-3}$) | Reference |
|---|---|---|---|---|
| Summit, Greenland | coastal | May-December | 0.3 | (von Schneidemesser et al., 2009) |
| Azores, Portugal | coastal | Winter | 6.6 | (Puxbaum et al., 2007) |
| Sonnblick, Austria | coastal | Winter | 12.4 | (Puxbaum et al., 2007) |
| Puy de Dôme, France | coastal | Winter | 18.3 | (Puxbaum et al., 2007) |
| Schauinsland, Germany | coastal | Winter | 33.7 | (Puxbaum et al., 2007) |
| Guangdong, China | forest | August | 25 (0.3-61) | (Wang et al., 2008) |
| Jilin, China | forest | July | 42 (32-67) | (Wang et al., 2008) |
| Hainan, China | forest | November | 107 (19-398) | (Wang et al., 2008) |
| Shanghai, China | forest | June | 143 (20-212) | (Wang et al., 2008) |
| Langley, Canada | rural | August | 26 | (Leithead et al., 2006) |
| Hok Tusi, Hong Kong | rural | Spring | 30 | (Sang et al., 2011) |
| Ovens, Australia | rural | Autumn | 870 | (Reisen et al., 2011) |
| Manjimup, Australia | rural | Autumn | 1060 | (Reisen et al., 2011) |
| Kowloon, Hong Kong | urban | Spring | 36 | (Sang et al., 2011) |
| National University, Singapore | urban | September | 91.2 | (Yang et al., 2013) |
| Shanghai, China | urban | Spring | 66 (18-159) | (Li et al., 2016) |
| Shanghai, China | urban | Summer | 28 (8.6-194) | (Li et al., 2016) |
| Shanghai, China | urban | Autumn | 229 (13-1606) | (Li et al., 2016) |
| Shanghai, China | urban | Winter | 161 (26-614) | (Li et al., 2016) |
| Southeastern US, USA | urban | Winter | 204.5 | (Zhang et al., 2010a) |
| Beijing, China | urban | Summer | 230 | (Cheng et al., 2013) |
| Beijing, China | urban | BB episode | 750 | (Cheng et al., 2013) |
| Beijing, China | urban | Typical summer | 120 | (Cheng et al., 2013) |
| Beijing, China | urban | Winter | 590 | (Cheng et al., 2013) |
| Beijing, China | urban | Firework episode | 460 | (Cheng et al., 2013) |
| Beijing, China | urban | Typical winter | 640 | (Cheng et al., 2013) |
| Grenoble, France | urban | January | 815 | (Favez et al., 2010) |
| Nanjing, China | urban | Winter | 373 (22.4-1476) | Current study |

**Anonymous Referee #2**

*The study quantitatively evaluated the influence of long-range transport of biomass burning (BB) emissions on carbonaceous aerosols in Nanjing city, China, which is helpful to understand the sources and associated contributions to aerosol carbonaceous component in urban regions in China. Following issues need to be addressed prior to publication.*

1. *One concern is about the relative contributions of long-range transport BB versus the local domestic BB emissions shown in Fig. 7. Though the levoglucosan concentration contained in air masses from southeast China is higher than that from the coastal region of eastern China, the air mass frequency from the latter air mass cluster (32.5%) is much higher than the former one (20%). After weighted with the cluster frequency, I would say the contribution from coastal eastern China actually is larger than that from southeast China. I understand that the long-range BB could contribute to aerosols in Nanjing significantly during certain time period, but, from long-term perspective, it might not be the case due to the large contribution from local domestic BB emissions. As such, the authors may want to make changes on the statements related to the contribution of long-range contribution.*

Reply: As suggested, we have revised the statements. The revised statements along with original statements in Page 12 line 7-9 make a more comprehensive discussion about the contribution of long-range transport and the potential impacts from domestic biomass burning emissions in this study. The statements after revision are as below, 'In addition, there was a notable cluster originating from the coastal site of eastern China and mainly passing through Shanghai and Jiangsu, where no fire signals were found. But this cluster had a relatively high mean levoglucosan concentration (673 ng m$^{-3}$). According to the research by Zhou et al. (2017) on the biomass burning emission inventory, domestic straw burning was responsible for ~42 % of total biomass burning in China in 2012, and ~1.5 Gg and ~80 Gg PM$_{2.5}$ came from domestic burning in Shanghai and Jiangsu, respectively. It illustrated the significant impact of indoor biomass burning which was invisible in MODIS fire spots map because satellite can only detect open biomass burning. Therefore, the levoglucosan enrichment of the cluster from coastal area might be due to domestic biomass burning. The air mass frequency of this cluster was 32.5 %, the largest one among that for all clusters, which means domestic biomass burning in the eastern China much influenced aerosols in Nanjing. Domestic biomass burning should be taken as an important factor to study aerosols of eastern China in further study, even though open biomass burning from long-range transport played a significant role in this study (Page 12 line 5-14).'

The original relative statements were 'Notably, there is a cluster originating from the coastal site of eastern China mainly passing through Shanghai and Jiangsu, where no fire signals are found. But this cluster has a relatively high mean levoglucosan concentration (673 ng m$^{-3}$). According to the research by Zhou et al. (Zhou et al., 2017) on biomass burning emission inventory, domestic straw burning was responsible for ~42 % of total biomass burning in China in 2012, and ~1.5 Gg and ~80 Gg PM$_{2.5}$ came from domestic burning in Shanghai and Jiangsu, respectively. It illustrated the significant impact of indoor biomass burning which was invisible in MODIS fire spots map. Therefore, the levoglucosan enrichment of the cluster from coastal area might be due to domestic biomass burning. Even if local domestic biomass burning proves to be an important contributor, the effects of long-distance transmission demonstrated above cannot be ignored'.

2. *The physical and optical properties of black carbon after emitted into atmosphere for certain time can be largely modified by the aging process (e.g., Peng et al., PNAS, 2016; Wang et al., JAMES, 2018). Please add some reviews regarding soot aging in the introduction section on page 4 when reviewing the alterations of physical and optical properties with the transport process.*

Reply: As suggested, we added review about soot aging in the introduction as 'Using a novel chamber method, Peng et al., (2016) found two stages in the aging of BC: initial transformation with little absorption variation and subsequent growth with a profound absorption enhancement. The fact was also revealed that BC under more polluted urban background had a more remarkable impact on the pollution development and the radiative forcing enhancement (Peng et al., 2016). Model simulations calibrated by chamber measurements showed significant differences in BC burden and radiative forcing from the simulations without considering BC aging. In addition, BC coating materials from aging processes were found responsible for a net increase of BC radiative forcing (Wang et al., 2018b) (Page 4 lines 21-27)'.

3. *In abstract: when talking about absorption of carbonaceous aerosols, the two terms (i.e., WSOC and water soluble BrC) are basically equivalent, but the authors switched back and forth to use them. Please stick to one expression.*

Reply: As suggested, WSOC has been replaced by water-soluble BrC when talking about the radiative absorption in abstract. The revised statements are 'Furthermore, the strong correlation (p < 0.01) between biomass burning tracers (such as lev) and light absorption coefficient ($b_{abs}$) for water-soluble brown carbon (BrC) revealed that biomass burning emissions played a significant role in the light-absorption properties of carbonaceous aerosols (Page 1 line 28-30)' and 'The solar energy absorption of water-soluble BrC due to biomass burning was estimated as $0.2 \pm 0.1$ (0.0-0.9) W m$^{-2}$, considering the biomass burning contribution to carbonaceous aerosols. Potential Source Contribution Function model simulations showed that the solar energy absorption induced by water-soluble BrC and EC aerosols was mostly due to the regional transported carbonaceous aerosols from the source regions such as southeastern China (Page 2 line 1-4)'.
The original statements were 'The solar energy absorption of WSOC due to biomass burning was estimated as $0.2 \pm 0.1$ (0.0-0.9) W m$^{-2}$. Potential Source Contribution Function model simulations showed that the solar energy absorption induced by WSOC and EC aerosols was mostly due to the regional transported carbonaceous aerosols from the source regions such as southeast China.'

4. *Since the observations are ground-based, while the back-trajectory analysis focuses on 500m level. Are the conclusions about relative contributions from different air mass clusters sensitive to the selected altitude?*

Reply: Considering that the planetary boundary layer usually has a lower height in wintertime, 500 m is a proper height set as the starting point altitude during the calculation of back-trajectory, which is universal in other studies (Yang et al., 2017;Rashki et al., 2015).

Yang, W., Wang, G., and Bi, C.: Analysis of Long-Range Transport Effects on PM2.5 during a Short Severe Haze in Beijing, China, Aerosol and Air Quality Research, 17, 1610-1622, 10.4209/aaqr.2016.06.0220, 2017.
Rashki, A., Kaskaoutis, D. G., Francois, P., Kosmopoulos, P. G., and Legrand, M.: Dust-storm dynamics over Sistan region, Iran: Seasonality, transport characteristics and affected areas, Aeolian Research, 16, 35-48, https://doi.org/10.1016/j.aeolia.2014.10.003, 2015.

*5. Typos Line 26 page 2: missing a preposition at ". . .frequently method biomass burning. . ."*

Reply: The statement has been revised as 'In the observation-based studies, the characteristic ratio and positive matrix factorization are frequently used to calculate biomass burning contribution (Page 2 line 29-30)'.

The original statement was 'In the observation-based studies, the characteristic ratio calculation and positive matrix factorization (PMF) are the most frequently methods biomass burning contribution.'

**Anonymous Referee #3**

*The manuscript measured the chemical and optical properties of carbonaceous aerosols in Nanjing and attempted to elaborate impacts of the biomass burning (BB) originated from regional transport on the carbonaceous aerosols. The topic is of interest but the manuscript is not well written. There are several factors hinder publication of the manuscript at the present form, and a major revision is needed.*

Major comments:

1. *The title is "Impact of regional-transported biomass burning emissions on chemical and optical properties of carbonaceous aerosols in Nanjing, East China". Therefore, It is assumed that the measured aerosol properties in Nanjing with and without BB impact should have large difference. Figure 5 shows that there are moments with low BB impact on Nanjing. Did the chemical and optical properties of carbonaceous aerosols under the low BB influence reveal a significant difference from those under the high BB influence?*

Reply: As suggested, T-test method has been applied to test the significance of chemical and optical properties difference between samples with high BB influence (BB-OC > 20.9%) and samples with low BB influence (BB-OC < 20.9%), where 20.9% is the mean BB-OC value of the event in this study. The T-test results shows that all of carbonaceous components (OC, EC, WSOC), BB tracers (lev, man, gal) and $b_{abs}$ of water-soluble BrC exhibit a significant difference between samples under high and low BB influence (h = 1, p < 0.05).

In addition, for a better representative of the study, we changed the title as "Chemical and Optical Properties of Carbonaceous Aerosols in Nanjing, East China: Regional-transported Biomass Burning Contribution (Page 1 line 1-3)".

The attached table shows the T-test results:

|  | OC | EC | WSOC | levoglucosan | mannanson | galactosan | $b_{abs}$ |
|---|---|---|---|---|---|---|---|
| h | 1 | 1 | 1 | 1 | 1 | 1 | 1 |
| sig | 0.0111 | 0.0083 | 0.0022 | 0.0000 | 0.0000 | 0.0000 | 0.0004 |

2. *Figure 5 shows that the fraction of BB-OC and BB-WSOC is nearly the same throughout the measurement. Does this mean that the emission ratios of WSOC/OC from other anthropogenic sources are the same as BB, or is it the coincidence from the calculation by the equation 6 and 7?*

Reply: We agree that the same trends and the good correlation of BB-OC and BB-WSOC in Figure 5 were due to the calculation coincidence.

However, even though the reference parameters for BB-OC and BB-WSOC calculation are from two separate studies, the BB-OC and BB-WSOC fraction are in a similar level with a linear regression equation slope close to 1 (0.96). It demonstrated the biomass burning source is clearly separated from other sources in our study.

The statements about BB-OC calculations are in Page 10 line 21-26, as 'The contribution of biomass burning to OC in PM2.5 (BB-OC) was calculated using Eq. (6):

$$BB - OC = \frac{(lev/(1000 \times OC))_{ambient}}{0.082} \times 100\ \%, lev: ng\ m^{-3}\ ; OC: \mu g\ m^{-3} \qquad (6)$$

(Zhang et al., 2010b; Puxbaum et al., 2007; Zdráhal et al., 2002). In biomass burning source emission tests for three major types of cereal straw (corn, wheat, and rice) in China, 0.082 was reported as the lev/OC ratio for $PM_{2.5}$ (Zhang et al., 2007). This value can be used in combination with the lev/OC ratios of our $PM_{2.5}$ samples to roughly estimate the contribution of biomass burning smoke to the ambient OC.'

The statements about BB-WSOC calculations are in Page 10 line 31-Page 11 line 2, as 'The contribution from biomass burning to the WSOC in PM2.5 (BB-WSOC) trend in Fig. 5 makes the above analysis more convincing due to the consistency of the contributions computed by two different methods. BB-WSOC was estimated with Eq. (7):

$$BB - WSOC = \frac{(lev/WSOC)_{ambient}}{0.17} \times 100\ \%, lev/WSOC : \mu g\ \mu gC^{-1} \qquad (7)$$

which used a lev/WSOC ratio of 0.17 $\mu g\ \mu gC^{-1}$ from the test burns of rice straws and wheat straw (Yan et al., 2015).'

The attached is Figure 5 (Page 30)

[Figure]

**Figure 5: Time series of biomass burning contribution to OC (BB-OC) and WSOC(BB-WSOC).**

3. *Since the authors attempted to investigate the impact of BB on carbonaceous aerosols, why did the authors only measure the absorption of BrC in WSOC, instead of the absorption of total BrC in OC? There are also water-insoluble BrC from BB.*

Reply: First, we admit that biomass burning releases water-insoluble BrC as well. But water-soluble BrC, which is our focus in this study, takes a great part of total BrC. According to our results, WSOC accounted for averaged 50% and up to 90% of OC; The light-absorbing property of water-soluble BrC ($b_{abs}$) showed a strong correlation with OC ($R^2 = 0.72$, p < 0.01). Additionally, WSOC has always been the research hotspot due to its effects on hygroscopicity of aerosols and the ability to serve as cloud condensation nuclei (CCN) (Novakov and Corrigan, 1996). Previous studies also showed that biomass burning was one of the predominant sources of WSOC and water-soluble BrC (Hoffer et al., 2006). Therefore, it's of the great significance to discuss the light-absorbing property of WSOC in particular. Nevertheless, we would like to classify total BrC according to the dissolvability in our future research and further explore the optical property of BrC.

Hoffer, A., Gelencsér, A., Guyon, P., Kiss, G., Schmid, O., Frank, G. P., Artaxo, P., and Andreae, M. O.: Optical properties of humic-like substances (HULIS) in biomass-burning aerosols, Atmos. Chem. Phys., 6, 3563-3570, 10.5194/acp-6-3563-2006, 2006.

Novakov, T., and Corrigan, C. E.: Cloud condensation nucleus activity of the organic component of biomass smoke particles, Geophysical Research Letters, 23, 2141-2144, 10.1029/96GL01971, 1996.

4. *Page 8, Line 17, the authors concluded that secondary OC is weak and stable only based on the OC/EC ratio ranging from 2 to 4 during the severe pollution period. The conclusion is arbitrary. Firstly, the OC/EC ratio ranging from 2 to 4 are rather large. Secondly, previous studies have indicated that SOA contribution during haze days during wintertime in China is considerable. For example, Huang et al. (2014) showed SOA contributed 44%-71% to the OA mass in four megacities of China during wintertime. Sun et al. (2013) revealed that SOA contributed 31% to the OA mass in Beijing in winter.*

Reply: As suggested, we revised the statements and added the references to describe the results more objectively, as 'Generally, an OC/EC ratio above two means a significant contribution of secondary organic aerosol (SOA) (Haque et al., 2019; Kunwar and Kawamura, 2014). As the pollution got more severe over time, it shrank to 2.0-4.0 after 16 Jan, indicating that the secondary sources had a relatively stable but still considerable impact on $PM_{2.5}$ at the polluted stage. According to the previous studies, SOA usually played an important role in the wintertime air pollution (Huang et al., 2014; Sun et al., 2013) (Page 8 line 22-26)'.

The original statements were 'In contrast to combustion sources, the secondary sources had a relatively weak and stable impact on $PM_{2.5}$ at the polluted stage.'

Minor comments:

5. *Page 2, Lines 11-15: What is the difference among those portions of OC?*

Reply: Each portion of OC has different size distributions, chemical molecular compositions or physical morphology (i.e. fractal or spherical) leading to the different optical properties. Here, we introduced a relative reference in the text (Page 2 line 15). According to the literature, this classification depends on the optical property of OC, as stated in text 'Meanwhile, a portion of organic carbon (OC) could yield a cooling effect on the land surface by scattering sunlight (Zhang et al., 2017; Myhre et al., 2013). A portion of OC involves in the aging process of elemental carbon (EC) or black carbon (BC) as coating materials for the BC core, enhancing BC radiative absorption (Peng et al., 2016; Wang et al., 2018b). A fraction of OC can also directly absorb solar energy functioning similarly as BC, which is referred to as brown carbon (BrC) (Laskin et al., 2015). But BrC is different from BC due to its strong light absorption from visible (VIS) to ultraviolet (UV) wavelengths, thus having a stronger wavelength dependence than BC (Andreae and Gelencsér, 2006) (Page 2 line 11-17)'.

6. *Page 2, Lines 15-16: Does it indicate that BrC is more absorptive than BC?*

Reply: It didn't indicate that BrC is more absorptive than BC. The sentence is revised as 'But BrC is different from BC due to its strong light absorption from visible (VIS) to ultraviolet (UV) wavelengths, thus having a stronger wavelength dependence than BC (Andreae and Gelencsér, 2006)' (Page 2 line 15-17), only describing the light absorbing characteristics of BrC which is different from BC. The estimated radiative forcing of BC and BrC in other literature were given following this sentence as 'Radiative forcing of BC is estimated to be 0.2-1.2 W $m^{-2}$ (Moffet and Prather, 2009), while radiative forcing of BrC contribution is suggested to be up to 0.25 W $m^{-2}$ or contribute up to 19 % of the total atmospheric absorption computed by model simulations (Feng et al., 2013) (Page 2 line 17-19)', showing that BC makes a stronger radiative forcing than BrC, but BrC absorption is non-negligible.

7. *Page 2, Line 26: what is "the most frequently method"?*

Reply: The statement has been revised to eliminate the expression ambiguity as '==In the observation-based studies, the characteristic ratio and positive matrix factorization are frequently used to calculate biomass burning contribution== (Page 2 line 29-30)'. Both the characteristic ratio calculation and positive matrix factorization are frequently methods to calculate the biomass burning contribution, which were applied in this study.

The original statements were 'In the observation-based studies, the characteristic ratio calculation and positive matrix factorization (PMF) are the most frequently methods biomass burning contribution.'

8. *Page 3, Line 3, "that" should be removed?*

Reply: It has been revised to as suggested, that '==These studies have demonstrated a significant importance of biomass burning contributions to carbonaceous aerosols== (Page 3 line 7-8)'.

9. *Page 3, Line 18, "impact" should read "impacts".*

Reply: It has been revised as suggested, that 'Biomass burning aerosol has significant ==impacts== on regional and global radiative forcing (Ramanathan and Carmichael, 2008) (Page 3 line 22)'.

10. *Page 3, Line 21: "than" should read "compared to".*

Reply: It has been revised as suggested, that '…has 70 % enhancement in biomass burning period ==compared to== normal days in Boulder, Colorado (Page 3 line 25)'.

11. *Page 3, Lines 25-27: Are the three parameters are specially used for BrC from BB?*

Reply: These three parameters are main parameters to describe optical properties of both BrC and EC from all kinds of sources. Research about biomass burning involving these parameters are listed afterwards, for our study focuses on the impact of biomass burning. We removed the misleading statements. The revised statements are '==Light-absorbing parameters including absorption coefficient, mass absorption efficiency and absorption Angstrom exponent, have been involved to describe the light absorption characteristics of BrC (Alexander et al., 2008; Yang et al., 2009; Chakrabarty et al., 2016; Choudhary et al., 2017; Srinivas et al., 2016). Hoffer et al. (2006) found that the contribution of humic-like substances to total aerosol absorption was as high as 50 % at the wavelength of 300 nm in Amazon basin. Hecobian et al. (2010) measured the optical properties of WSOC in Atlanta, US and their results showed that the absorption coefficient around 365 nm had a strong correlation with WSOC from biomass burning or from urban emissions. Du et al. (2014b) calculated mass absorbance efficiency for WSOC from a biomass burning factor derived from positive matrix factorization receptor model. It showed that the biomass burning source contributed to 58 % of total light absorption at 365 nm and WSOC associated with sulfate and oxalate contributed to 21 %== (Page 3 line 29-Page 4 line 2).'

12. *Page 3, Lines 28: What is HULIS?*

Reply: HULIS has been replaced by its full name, 'humic-like substances'. The sentence has been revised as 'Hoffer et al. (2006) found that the contribution of humic-like substances to total aerosol absorption was as high as 50 % at the wavelength of 300 nm in Amazon basin (Page 3 line 32-33).'

The original sentence was 'Hoffer et al. (2006) investigated the optics of humic-like substances (HULIS) in Amazon basin and found that the contribution of HULIS to total aerosol absorption was as high as 50 % at the wavelength of 300 nm.'

*13. Page 7, Line5, "give" should be "given".*

Reply: It has been revised as suggested, as 'The stand-alone RRTMG_SW can be driven by a given atmospheric profile (Page 7 line 12)'.

*14. Page 8, Line 10, "uncertainty" should be "variation"*

Reply: It has been revised as suggested, as 'The relative standard deviation was used to represent the variation of OC (44 %) and EC (53 %) … (Page 8 line 16-17)'.

*15. Page 10, Line 25, "which illustrating the necessity of parameter BB-OC". The sentence is hard to understand.*

Reply: We removed the sentences after Page 10 line 30, which were not closely related to the main idea of this paragraph.

The removed sentences are 'There were three remarkable BB-OC peaks at 08:00 on 20 Jan, 14:00 on 21 Jan and 08:00 on 25 Jan in Fig. 5, indicating the large contributions of biomass burning during those periods. However, EC and biomass burning tracers were not rich at 08:00 on 20 Jan, implying that combustion activities were not highly intense at that time. But it doesn't contradict the fact that biomass burning made a significant contribution to OC compared to other pollution sources at 08:00 on 20 Jan. It demonstrates that it's very necessary to use BB-OC to depict the contribution of biomass burning instead of biomass burning tracer levels, for BB-OC excludes the influence of the OC level.'

[revised manuscript text omitted]